# Viral proteogenomic and expression profiling during productive replication of a skin-tropic herpesvirus in the natural host

**Jeremy D. Volkening**[1], **Stephen J. Spatz**[2], **Nagendraprabhu Ponnuraj**[3], **Haji Akbar**[3], **Justine V. Arrington**[4], **Widaliz Vega-Rodriguez**[3], **Keith W. Jarosinski**[3]*

**1** BASE2BIO, Oshkosh, Wisconsin, United States of America, **2** US National Poultry Research Laboratory, ARS, USDA, Athens, Georgia, United States of America, **3** Department of Pathobiology, College of Veterinary Medicine, University of Illinois at Urbana-Champaign, Urbana, Illinois, United States of America, **4** Protein Sciences Facility, Roy J. Carver Biotechnology Center, University of Illinois Urbana-Champaign, Urbana, Illinois, United States of America

* kj4@illinois.edu

## Abstract

Efficient transmission of herpesviruses is essential for dissemination in host populations; however, little is known about the viral genes that mediate transmission, mostly due to a lack of natural virus-host model systems. Marek's disease is a devastating herpesviral disease of chickens caused by Marek's disease virus (MDV) and an excellent natural model to study skin-tropic herpesviruses and transmission. Like varicella zoster virus that causes chicken pox in humans, the only site where infectious cell-free MD virions are efficiently produced is in epithelial skin cells, a requirement for host-to-host transmission. Here, we enriched for heavily infected feather follicle epithelial skin cells of live chickens to measure both viral transcription and protein expression using combined short- and long-read RNA sequencing and LC/MS-MS bottom-up proteomics. Enrichment produced a previously unseen breadth and depth of viral peptide sequencing. We confirmed protein translation for 84 viral genes at high confidence (1% FDR) and correlated relative protein abundance with RNA expression levels. Using a proteogenomic approach, we confirmed translation of most well-characterized spliced viral transcripts and identified a novel, abundant isoform of the 14 kDa transcript family via IsoSeq transcripts, short-read intron-spanning sequencing reads, and a high-quality junction-spanning peptide identification. We identified peptides representing alternative start codon usage in several genes and putative novel microORFs at the 5' ends of two core herpesviral genes, pUL47 and ICP4, along with strong evidence of independent transcription and translation of the capsid scaffold protein pUL26.5. Using a natural animal host model system to examine viral gene expression provides a robust, efficient, and meaningful way of validating results gathered from cell culture systems.

**Data Availability Statement:** The LC-MS data have been deposited with the ProteomeXchange Consortium via the jPOST partner repository

(https://jpostdb.org) with the dataset identifier PXD040077. Raw transcriptome sequencing data is available in NCBI under BioProject PRJNA934312. Short-read under SRA accessions SRR23440507 - SRR23440518 and long-read BioSample accessions SAMN35174587 - SAMN35174589 and SRA accessions SRR24653946 - SRR24653948.

**Funding:** This report was supported by Agriculture and Food Research Initiative Competitive Grant nos. 2013-67015-26787, 2016-67015-26777, and 2020-67015-21399 from the USDA National Institute of Food and Agriculture to KWJ, and USDA-ARS NACA agreements nos. 58-6040-8-037 and 58-6040-0-015 to KWJ and SJS. The funders had no role in study design, data collection and analysis, decision to publish, or preparation of the manuscript.

**Competing interests:** The authors have declared that no competing interests exist.

## Author summary

In the natural host, the transcriptome and proteome of many herpesviruses are poorly defined. Here, we evaluated the viral transcriptome and proteome in feather follicle epithelial skin cells of chickens infected with Marek's disease virus (MDV), an important poultry pathogen as well as an excellent model for skin-tropic human alphaherpesvirus replication in skin cells. Using fluorescently tagged virus, we significantly enriched the number of infected cells sampled from live chickens, greatly enhancing the detection of viral transcripts and proteins within a host background. We carefully defined the *in vivo* transcriptome using both short-read RNA-Seq and long-read full-length transcript sequencing, and deep MS/MS-based proteomics from the enriched virus-infected feather follicle samples confirmed the translation of most transcripts and identified novel expressed peptides supportive of an increasingly complex translational and regulatory viral landscape. The demonstrated deep peptide sequencing capability can serve as a template for future *in vivo* work in herpesviral proteomics.

## Introduction

Herpesviruses have two modes of spread, cell-to-cell and cell-free virion release, where the significant advantage of cell-to-cell spread is evasion of the immune system. However, infectious cell-free virus must be released into the environment to disseminate amongst a population [1]. Herpesviruses are highly adapted to their host species, having co-evolved for millions of years, which makes studying natural virus transmission in the host population difficult, and for humans nearly impossible. In addition, most herpesviruses are primarily cell-associated within the host where virions are delivered through cell-cell junctions and tunneling nanotubes [2], likely to evade the host immune system [3]. Depending on the herpesvirus, the amount of infectious cell-free virus released from cells in cell culture is highly variable. Most studies have focused on *in vitro* cell culture models to study herpesvirus replication that is primarily by cell-to-cell spread. As the transcriptional and translational machinery active during the cell-associated and cell-free stages of the viral life cycle is likely to vary significantly, we sought to address expression of viral genes during replication in a natural herpesvirus animal model system.

Marek's disease virus (MDV; *Gallid alphaherpesvirus 2*; GaHV-2) is a significant pathogen affecting the poultry industry due to its global distribution and transmissibility. MDV is a member of the *Herpesviridae*, subfamily *Alphaherpesvirinae*, and is related to the human herpes simplex virus 1 (HSV-1), HSV-2, and varicella-zoster virus (VZV), both genetically [4,5] and, in particular, in their shared tropism to epithelial skin cells required for replication, egress, and dissemination into the environment. In contrast to other well-studied alphaherpesviruses, MDV is strictly cell-associated when grown in cell culture, relying on spread through cell-to-cell contact. To date, no cell-free virions have been produced using primary cell culture or engineered immortalized continuous cell lines [6–11]. The only cells known to produce infectious cell-free virus are differentiated chicken epithelial skin cells called the feather follicle epithelium [12]. The production of cell-free virus is required for host-to-host transmission, and specific viral genes required for this process have been identified [13]. The related human VZV is also primarily cell-associated in cell culture [14], with only small amounts of infectious cell-free virus produced, while the prototype alphaherpesvirus HSV-1 generates cell-free virus that is partially dependent on the cell type [15]. However, for these skin-tropic viruses, human-to-human transmission cannot be studied, and mouse models do not facilitate

transmission. Thus, the MDV-chicken model system is well suited to address transmission and production of cell-free virus in the host.

The ~180 kb double-stranded DNA genome of MDV was first sequenced in 2000 for the very virulent Md5 [16] and attenuated GA [17] strains, and was annotated to have 338 open reading frames (ORFs) of >60 aa in length, of which 103 ORFs were predicted to be functional. The current annotation of the MDV genome largely relies on both *in silico* ORF predictions and homologous ORFs in related alphaherpesviruses [18]. Recently, studies on the MDV transcriptome have been reported in cell culture [19] and *in vitro* infected B cells [20], expanding our knowledge of MDV's complex gene expression patterns in different types of cells. In addition, the transcriptome of MDV-infected feathers in chickens has been recently reported, with limited success in identifying viral transcripts deemed necessary for productive infection [21]. Mass spectrometry (MS)-based proteomics studies have also provided useful information on viral proteins produced in MDV-transformed chicken cells [22] and during *in vitro* replication in cell culture [23] and B cells [20]. Together, these studies have provided a foundational understanding of viral transcription and translation, but they are limited either by an *in vitro* context or a shallow breadth and depth of coverage.

Over the past decade, we have established a robust natural infection system by which we can identify and enrich MDV-infected epithelial skin cells from live chickens using fluorescence microscopy without complex manipulation of the samples [24–26]. To further our knowledge of the viral machinery active during the critical stages of virus assembly, egress, and shedding, we herein combined this system with short- and long-read RNA sequencing as well as bottom-up proteomics of the MDV-enriched samples to define the combined viral transcriptome and proteome during replication within the natural host, achieving deep coverage at both levels and uncovering novel aspects of *in vivo* mRNA expression, splicing, translation, and promoters usage in epithelial skin cells.

## Results & discussion

### Visualization of the transcriptional and translational landscape in epithelial skin cells

The overall experimental design is shown in Fig 1 and described in the Materials and Methods. Briefly, in experiment 1, mock and MDV-infected samples were processed for short-read RNA sequencing and LC-MS/MS analyses. In experiment 2, mock and MDV-infected samples were processed for long-read RNA sequencing. Overviews of strand-specific short read coverage, RNA-Seq-based splicing events detected, IsoSeq transcript models, and MS/MS peptides are shown for the unique long (Figs 2 and 3), repeat long (Fig 4), repeat short (Fig 5), and unique short (Fig 6) regions. These genome tracks and additional data tracks generated from this study can also be viewed interactively at https://igv.base2.bio/AAG3-9Fja-99a2-2asZ/, and the track files in BED, GFF3, and bedgraph formats are included in the S1 File.

### The viral proteome in epithelial skin cells in the host

LC-MS/MS-based bottom-up proteomics was used to examine the expressed proteome of MDV in epithelial skin cells. Both total protein and phospho-enriched samples were used to increase the coverage of total viral proteins (Fig 1). Each of the six replicates (3 infected, 3 mock-infected) produced 105k-122k total MS2 spectra from the unenriched samples and 68k-88k MS2 spectra from the phospho-enriched fractions (S1 Table). Spectra matched to peptides (peptide-spectrum matches; PSMs) ranged from 13k-23k for unenriched samples and 4k-11k for phospho-enriched samples [1% PSM false discovery rate (FDR)]. Of these PSMs, the

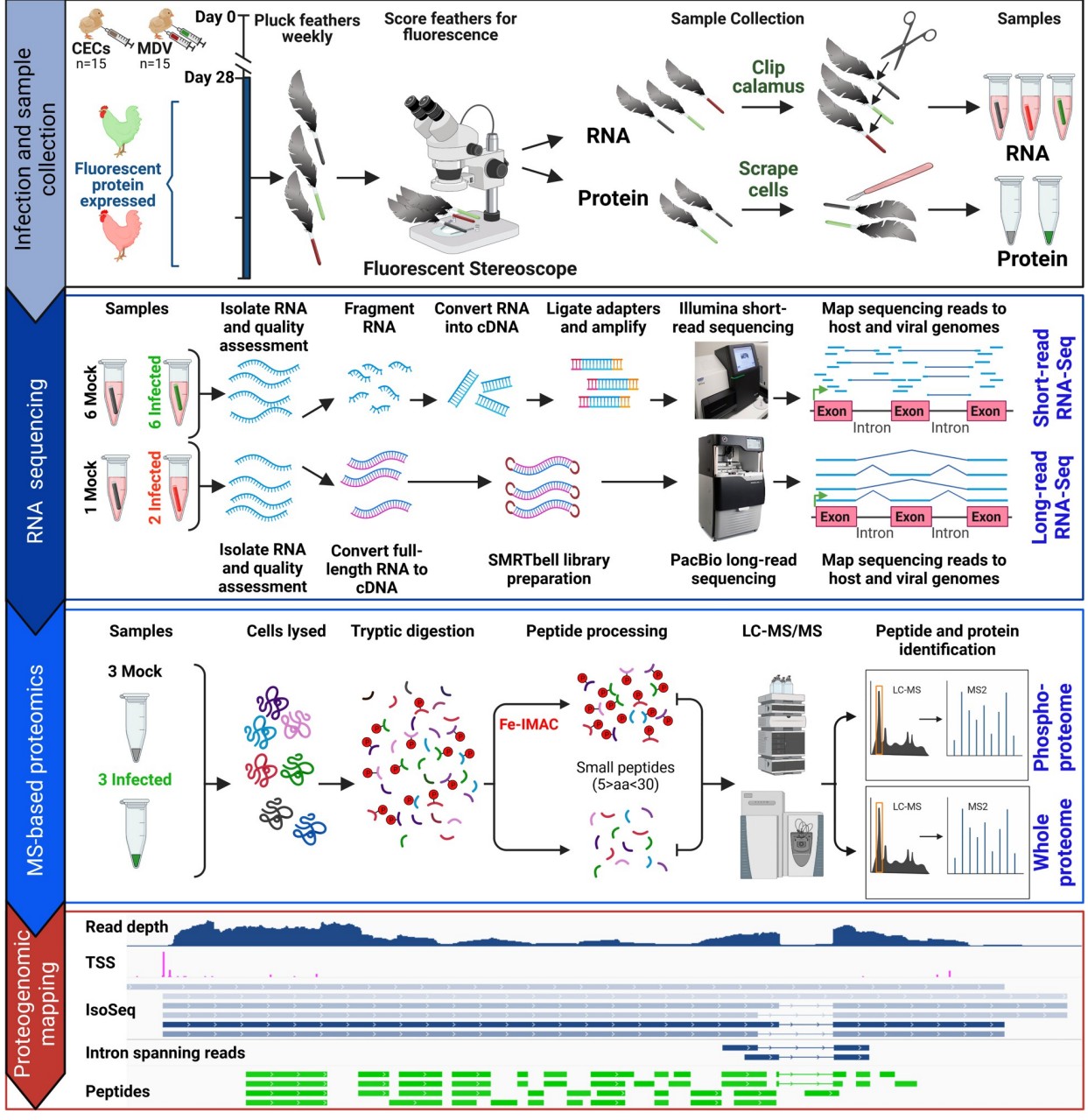

**Fig 1. Schematic illustration of the experimental approach.** Three-day old chicks were experimentally infected with MDV or mock-infected. Feathers were plucked weekly and birds heavily infected in feathers were sacrificed for sample collection, along with age-matched controls. Samples were collected between 21 and 35 days based on the level of pUL47eGFP or RLORF4mRFP. Approximately 6–10 feathers per bird were directly clipped at the calamus, dropped in ice-cold RNA STAT60, snap frozen on dry ice, and stored at -80˚C until processed for RNA extraction and subsequently RNA sequencing (see Materials and Methods). Approximately 3–4 feathers per bird (n = 3/group) were used to collect protein by scraping fluorescent cells into an Eppendorf tube, and all samples were stored at -80˚C until processed for MS-based proteomics (see Materials and Methods). RNA sequencing and LC-MS/MS were performed on samples and used for proteogenomic mapping in epithelial skin cells. Image created with BioRender.com.

fraction matching MDV proteins ranged from 5.6–6.1% for infected unenriched samples and 8.7–9.4% for infected phospho-enriched samples. Total MDV-matched PSM counts in mock-infected replicates were 0 or 1 for both unenriched and phospho-enriched fractions, suggestive

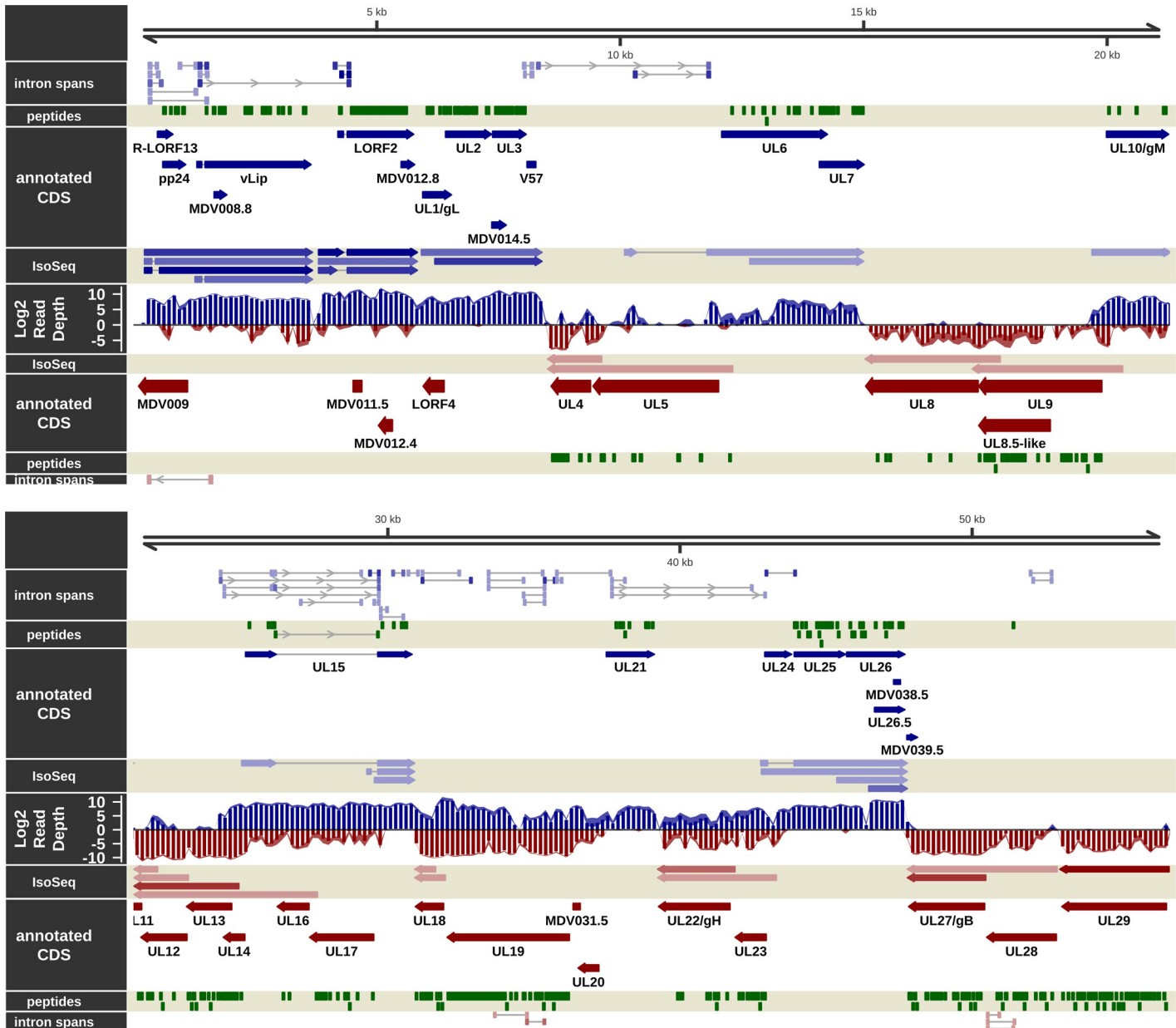

**Fig 2. The transcriptome and proteome of the unique long (UL) region of MDV in epithelial skin cells, part 1.** Visualization of the RNA-Seq-supported intron spans, annotated coding sequence, IsoSeq transcripts, log2 short read depth, and peptides detected on the forward (blue) and complementary (red) strands within the UL region plus 800 bp upstream flanking sequence of MDV from MDV008 to MDV042. A subset of calculated IsoSeq transcripts were selected for display to show primary transcripts for each gene as well as abundant alternative isoforms. Estimated background read depth (116) was subtracted from RNA-Seq coverage plots before log2 transformation and display.

of a very high specificity in peptide assignment. The MDV PSMs represented 1,484 distinct annotated viral peptides identified at 1% peptide FDR, not including different isoforms of each peptide such as post-translational modifications (PTMs) and missed cleavages; the full list is in S2 Table.

A total of 84 non-redundant MDV proteins (excluding terminal repeat copies but including different splice forms) were identified by at least one peptide at a maximum protein q-value of 0.01 (S3 Table). Of these, 79 proteins were identified by at least two distinct peptides at 1%

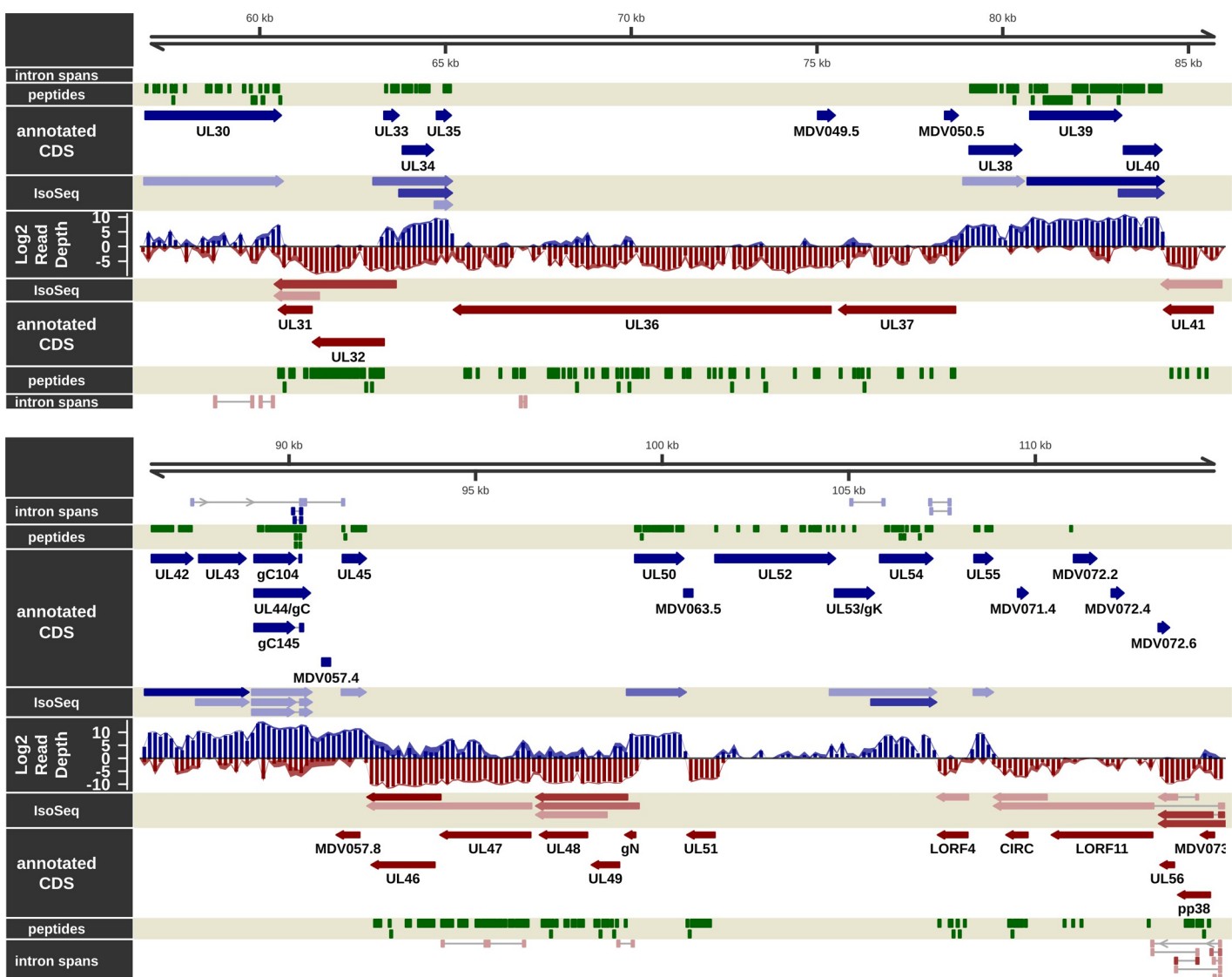

**Fig 3. The transcriptome and proteome of the unique long (UL) region of MDV in epithelial skin cells, part 2.** Visualization of the RNA-Seq-supported intron spans, annotated coding sequence, IsoSeq transcripts, log2 short read depth, and peptides detected on the forward (blue) and complementary (red) strands within the UL region plus 800 bp downstream flanking sequence of MDV from MDV043 to MDV073. A subset of calculated IsoSeq transcripts were selected for display to show primary transcripts for each gene as well as abundant alternative isoforms. Estimated background read depth (116) was subtracted from RNA-Seq coverage plots before log2 transformation and display.

FDR (commonly used criteria for protein presence), and 80 of the 84 proteins were detected in all three biological replicates. When considering expected peptide coverage (defined here as the protein length covered by detected peptides as a fraction of the total residues found in theoretical tryptic peptides $\geq$ 6 aa), 47 proteins had a breadth of coverage > 50%, 18 proteins had coverage > 80%, and five were over 90% covered. Histograms of unique peptide counts and breadth of coverage for detected viral proteins are shown in S1 Fig. Based on relative iBAQ (sum of peptide precursor intensities divided by theoretically observable peptides and divided again by the sum of all iBAQs), the most abundant protein present in epithelia skin cells was glycoprotein C (gC), followed by UL45 (envelope protein), UL42 (DNA Pol accessory), UL39 and UL40 (ribonucleotide reductase subunits), UL50 (Deoxyuridine 5'-

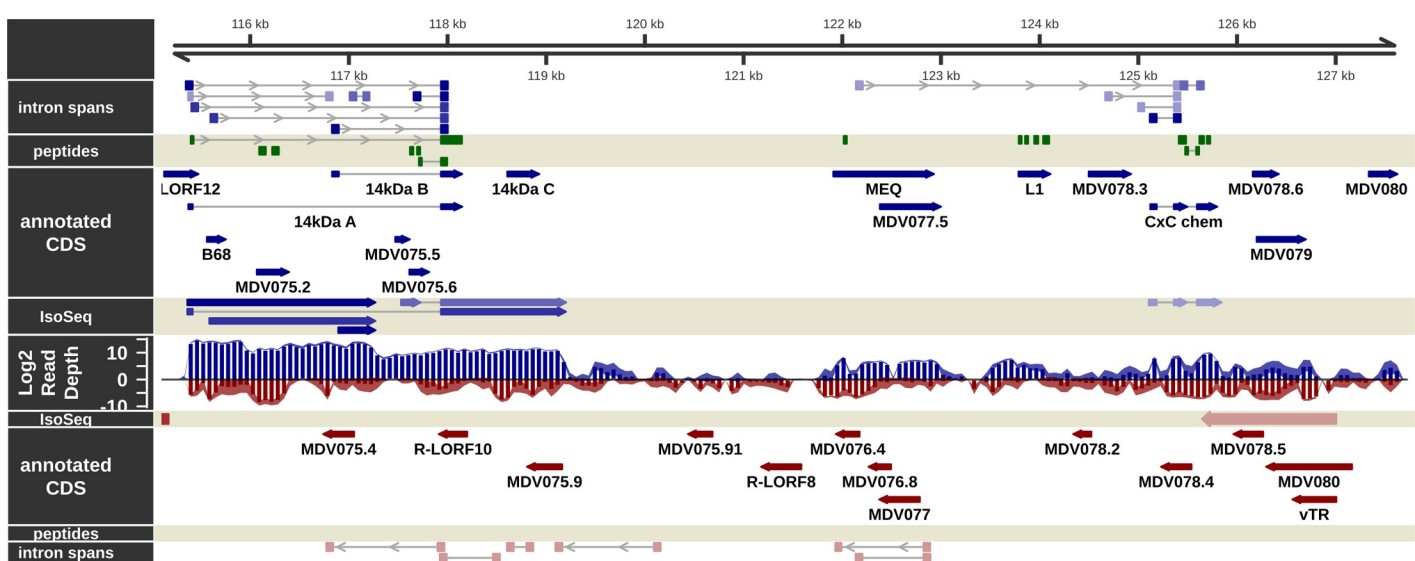

**Fig 4. The transcriptome and proteome of the internal repeat long (IRL) region of MDV in epithelial skin cells.** Visualization of the RNA-Seq-supported intron spans, annotated coding sequence, IsoSeq transcripts, log2 short read depth, and peptides detected on the forward (blue) and complementary (red) strands within the RL region of MDV from MDV074 to MDV080. A subset of calculated IsoSeq transcripts were selected for display to show primary transcripts for each gene as well as abundant alternative isoforms. Estimated background read depth (116) was subtracted from RNA-Seq coverage plots before log2 transformation and display.

triphosphate nucleotidohydrolase; DUT), UL49 (tegument protein VP22), and UL18 and UL19 (capsid subunits) (S3 Table). These nine proteins comprised an estimated 76% of the quantified viral protein load by molarity.

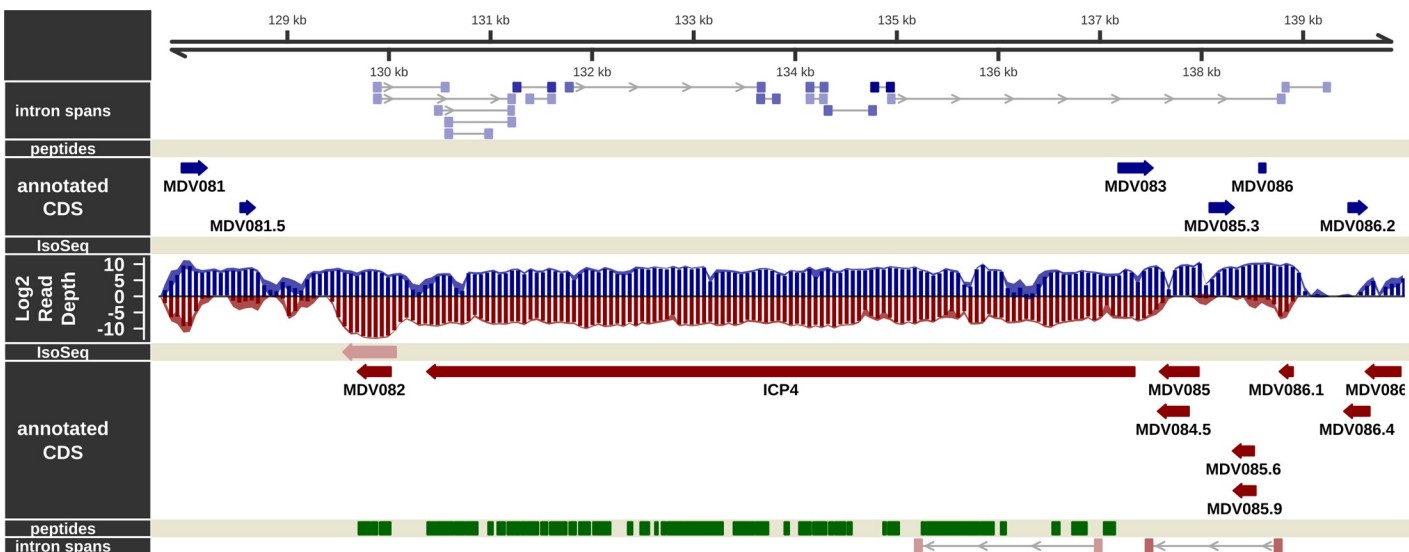

**Fig 5. The transcriptome and proteome of the internal repeat short (IRS) region of MDV in epithelial skin cells.** Visualization of the RNA-Seq-supported intron spans, annotated coding sequence, IsoSeq transcripts, log2 short read depth, and peptides detected on the forward (blue) and complementary (red) strands within the RS region of MDV from MDV081 to MDV086. A subset of calculated IsoSeq transcripts were selected for display to show primary transcripts for each gene as well as abundant alternative isoforms. Estimated background read depth (116) was subtracted from RNA-Seq coverage plots before log2 transformation and display.

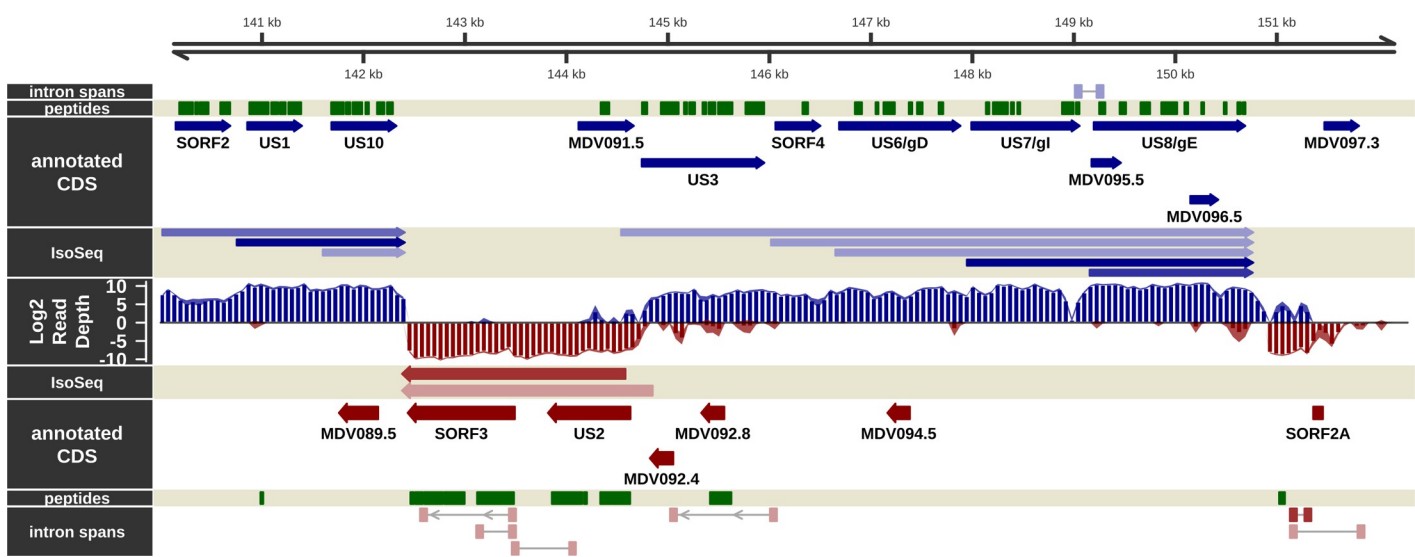

**Fig 6. The transcriptome and proteome of the unique short (US) region of MDV in epithelial skin cells.** Visualization of the RNA-Seq-supported intron spans, annotated coding sequence, IsoSeq transcripts, log2 short read depth, and peptides detected on the forward (blue) and complementary (red) strands within the US region plus 800 bp downstream flanking sequence of MDV from MDV086 to MDV097.3. A subset of calculated IsoSeq transcripts were selected for display to show primary transcripts for each gene as well as abundant alternative isoforms. Estimated background read depth (116) was subtracted from RNA-Seq coverage plots before log2 transformation and display.

## Single-peptide proteins

Using a single peptide as evidence of protein expression is generally unreliable, as even high-scoring PSMs can be incorrect due to factors such as an incomplete search database or unsearched PTMs. However, manual analysis can help to provide additional support either for or against the peptide match and confidence in the presence of the matched protein. Here, we used visual inspection of the matching MS2 b/y ion series and the extracted ion chromatogram (XIC) for the peptide mass and retention time to further evaluate peptides from single-peptide proteins. The use of the XIC by replicate was informative here because no elution peak would be expected in mock-infected replicates.

Protein UL11 (pUL11-CEP3) was identified from a single peptide in both phosphorylated and unphosphorylated forms in all three infected replicates. pUL11 is a short protein with only five predicted tryptic peptides ≥ 6 aa. Inspection of the MS2 spectra from three selected top PSMs of the unphosphorylated peptide shows a nearly complete y-ion series within the detectable m/z range and with no precursor co-isolation (S2 Fig). Additionally, elution peaks in the XIC plot are found only for the three infected replicates. This supplementary evidence provides strong support for identifying this peptide and thus, the expression of pUL11 in the skin cells during productive replication. A single (different) peptide from pUL11 was formerly identified in MDV-infected CECs [23], and it should be noted that most studies using MS-based proteomics do not typically detect this protein with efficiency [20,27,28]. For pUL49.5 or glycoprotein N (MDV064), inspecting the MS2 spectra and XIC for the single identified peptide provides similarly strong support for its identification and presence (S3 Fig). However, the MS2 spectra and XICs for the remaining three single-peptide proteins (MDV076/MEQ: SHDIPNSPS[+80]K; MDV093/SORF4: SRDFS[+80]WQNLNSHGNSGLR; MDV091.5: TINESLVPANPVPRT[+80]PVPSGGFVLTIGR), are less convincing, with less-complete b/y ion series and noisier XIC peaks (S4 Fig) that neither strongly support nor fully dispute the identifications.

## The MDV expressed transcriptome of epithelial skin cells

A summary of short-read RNA-Seq data for this experiment is shown in S4 Table. Library sizes for the twelve replicates ranged from $9.4\times10^6$ to $1.3\times10^7$ read pairs after trimming/filtering. The fraction of reads mapping to the MDV genome for infected samples ranged from 4.6% to 12.2%. For mock-infected replicates, the fraction of MDV-mapped reads was negligible (from 0 to 13 *total* mapped read pairs)–as with MS/MS, identification of MDV reads was highly specific to infected samples.

The libraries' overall strand specificity (number of read pairs mapping to the expected strand as a fraction of all classified read pairs) was high for the host genome (range 95–98%). However, it was observed that the strand specificity of viral reads was significantly lower (range 84–88%). For reference, the expected specificity for randomly distributed (i.e., not strand-specific) libraries would be 50%. From these values and visual inspection of the read alignments in IGV, there appeared to be a significant fraction of viral reads mapping to both presumed intergenic and presumed anti-sense regions of the viral genome. Visualization of host read alignments, on the other hand, appeared highly strand-specific and highly intragenic (S5 Fig). Because of this apparent background noise, possibly due to viral gDNA contamination, it was decided that traditional count-based or k-mer based methods of gene expression analysis would be unsuitable. Instead, the median read depth was calculated for each viral gene interval and replicate, as described in the Materials and Methods, and summarized to an average median read depth and standard deviation across all six replicates. Similarly, a baseline read depth distribution was calculated for intergenic/antisense regions (S6 Fig). A median depth threshold of two standard deviations above the mean background coverage (50-fold + 2 × 33-fold = 116-fold) was then used as the threshold for the "expression" of a gene with ~98% one-tailed confidence. This relatively conservative criterion was chosen in order that the results might be discussed and referenced as a high-confidence baseline for epithelial skin cells. Genes not passing this threshold may be expressed at low levels, but their presence and any biological significance of such low levels of expression would require further investigation. Approximately 75% (114/152) of non-redundant viral genes found in the RB-1B annotations used here were detected as expressed at the mRNA level above this threshold. Of the 152 gene models in the annotations used herein, 55 are annotated as "hypothetical proteins," likely based on *in silico* ORF prediction alone, and many overlap core or well-characterized genes. When these are ignored, we find evidence for the expression of 89/97 (92%) of the remaining annotated non-redundant genes in epithelial skin cells.

The most highly expressed transcript was MDV075.1/B68 (S3 Table). However, there was no evidence of the translation of this coding sequence in the MS/MS dataset. The expression values are possibly a result of non-spliced mRNA from the overlapping, and also highly expressed, 14 kDa protein gene family. The B68 ORF lies within the intron of MDV075 (14 kDa A), downstream of the first intron donor site (Fig 7A). The representative unspliced Iso-Seq transcript (discussed below) encompassing B68 also contains the downstream hypothetical ORF currently annotated as MDV075.2, and the CPC coding potential calculator (v2.0) [29] classified the transcript as non-coding, with coding probabilities of 0.05 and 0.19 for the two ORFs, respectively. Of note, the next most highly expressed gene was MDV082 (S3 Table), located in the short-inverted repeat (IRS) downstream of ICP4. Little was known about this gene until recently where it was found to be expressed late in the viral life cycle and enhanced the rate of disease progression, but it was not essential for replication, spread, or tumor formation [30]. We also found evidence of abundant expression of MDV082 at the protein level (~ 1% relative molar abundance, 92% protein coverage) and a well-supported full-length IsoSeq

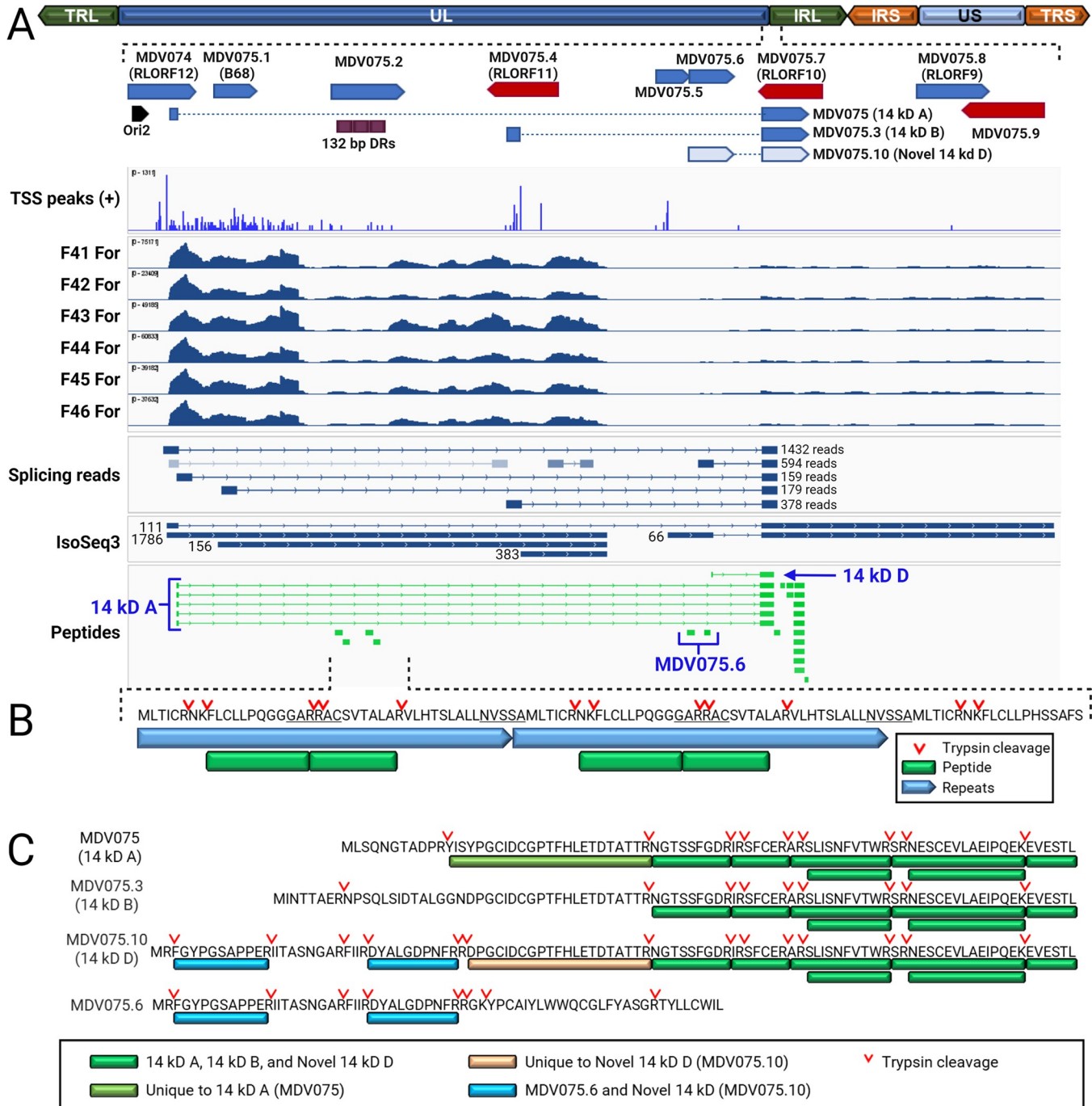

**Fig 7. Expression of the 1.8 kb family transcripts, novel mRNA splicing, and validation of protein expression in epithelial skin cells.** (A) Schematic representation of the MDV genome and location of the terminal (TRL) and internal (IRL) repeat long, unique long (UL), terminal (TRS) and internal (IRS) repeat short, and unique short (US) regions. The region encoding the 1.8 kb family transcripts is expanded from the IRL. A summary of transcripts detected or not detected in RNA sequencing and proteomics are shown along with the region of the 132 bp direct repeats. (B) Validation of the 132 bp direct repeat ORF encoding the MDV006.5 MDV075.2 protein. Peptides detected are noted in the figure legend along with tryptic cleavage sites. (C) Protein alignment of 14 kDa A, 14 kDa B, and Novel 14 kDa D. Peptides unique to each protein are noted in the figure legend along with tryptic cleavage sites. Image created with BioRender.com.

transcript model with a strong transcription start site peak (Fig 5). MDV057 (UL44-gC) transcripts were similarly highly abundant, mirroring their high abundance at the protein level.

Of the annotated coding sequences with read depths below the threshold level, most (30/38) are ORFs annotated as "hypothetical protein" and are unlikely to be functional in epithelial skin cells, including MDV013.5 (LORF4), MDV057.8 (LORF8; 23 kDa protein), MDV074 (RLORF12), MDV075.7 (RLORF10), and MDV077 (23 kDa nuclear protein). Of these genes, only MDV072 (LORF11) has evidence of expression at the protein level (four distinct peptides observed, q = 0.0004). Short-read median coverage for this gene was just below our threshold level for expression, but an IsoSeq transcript was detected with reasonable coverage (26 reads) and a spliced 5' UTR which is discussed later. Core herpesviral genes in this category included MDV017 (UL5) and MDV066 (UL52) which are both helicase-primase subunits. Both were detected with high confidence at the protein level, but, along with LORF11, they are the least abundant proteins present by estimated molarity (riBAQ) (S3 Table). MDV017 (UL5) was detected at low levels by IsoSeq but MDV066 (UL52) was not.

Overall, the correlation between RNA and protein levels was moderate, with Pearson R for log2 median read coverage vs. log2 riBAQ protein abundance = 0.62 (S7 Fig). This relatively low level of correlation could be the result of the complex post-transcriptional regulation of viral transcripts and translation of proteins. However, low levels of RNA/protein correlation are frequently observed due to differences in transcript vs. protein stability as well as measurement error.

## Long read transcript discovery

Because of the dense coding landscape of herpesviral genomes and the co-terminal nature of many gene clusters, interpretation of short-read RNA-Seq and assignment of short reads to specific genes can be challenging. Long-read sequencing technologies such as Oxford Nanopore and PacBio can address this issue by sequencing full-length cDNA molecules, allowing one to differentiate between overlapping transcripts with different transcription start sites, terminal cleavage sites, and splicing patterns. Here, PacBio IsoSeq sequencing was used to generate consensus transcript models from full-length, non-concatemeric (FLNC) reads for two infected replicates as well as one mock-infected control. Sequencing yields for the three samples ranged from 274k-310k FLNC reads; from 99.5–99.6% mapped to the combined chicken/viral reference; and the fractions mapping to the viral genome were 2.1% and 3.0% for the two infected samples (S5 Table). Although the fraction of viral reads was somewhat lower than for short-read RNA-Seq, SQANTI3 analysis suggested that sufficient depth was reached to approach saturation of the transcriptome, at least for primary transcripts (S8 Fig).

After discarding transcripts with summed read support < 4 and SQANTI3 filtering of probable genome-primed artifacts (poly-dT primers binding to A-rich genomic regions), 300 unique transcript models remained. Many of these represent low-count transcripts that differ from higher-abundance transcripts by alternate 5' start sites. This is due to the conservative settings used during transcript collapsing–because of the co-terminal nature of many herpesvirus' transcripts coding for different proteins, the default software settings to collapse all shorter transcripts that vary only by 5' location into the longest transcript would result in discarding many real, major transcripts. Therefore, we disabled this setting and adjusted the threshold 5' distance for collapsing transcripts (see methods). The trade-off was an increase in low-abundance, alternate transcripts, many of which may result from partial sequencing or rare transcription events. Of the 300 transcripts in the filtered set, 154 had combined supporting read counts ≥ 10 and 98 had read counts ≥ 20 (S9 Fig). The most abundant transcript had a combined supporting read count of 1786 (PB.1656.215; replicate C1: 756 reads; replicate D1: 1030

reads). This is an unspliced transcript with B68 as the 5'-most annotated coding sequence, in agreement with the short read data in which this was also the most abundant transcript (Fig 7B). It shares a transcription start site with the spliced 14kDa A transcript, which was also found in the transcript build (PB.16561.229; 111 total reads). As noted earlier, there was no evidence in the proteomics data that the B68 coding sequence is translated. However, both short- and long-read results suggest it is highly abundant in an unspliced form.

This transcript overlaps the variable 132 bp direct repeat region of MDV, and insertions of exact multiples of 132 bp are seen in many of the mapped FLNC reads at this locus (see IGV browser). The reference genome sequence used contains three copies of the direct repeat. When the insertions and deletions in FLNC reads overlapping the repeat region were tallied, considering only indels within 1nt of an exact multiple of 132 bp based on CIGAR string, 37% of reads differ in repeat copy number from the reference in replicate C1, and in replicate D1, 81% of overlapping reads differ in copy number from the reference, with 4 copies being most common (S10 Fig). Repeat copy numbers as high as 11 were observed. Given that both birds were inoculated from the same viral stock, these results suggest that the number of direct repeats is highly dynamic within the viral genome *in vivo*, and that this variability is expressed within the most abundant transcript in epithelial skin cells. Since it has been thought that expansion of the 132 bp repeat at the genomic level was likely the result of passaging in cell culture [31], it is interesting that our data show there appears to be fluctuation *in vivo* as well. The functional significance of this, if any, is not currently known.

## High-resolution transcription start site (TSS) mapping

Therefore, a separate analysis of precise TSS locations was performed based on mapping locations of FLNC reads. By calculating frequencies of FLNC read starts on the genome, we observed distinct peaks upstream of most annotated and well-characterized coding sequences (see S11 Fig for a representative example). Using these frequencies as input to a modified version of the previously described FocusTSS algorithm (see Materials and Methods), a set of 180 candidate TSS was produced (S6 Table). The candidates were manually curated to assign one major TSS to as many annotated genes as possible. Overall, 73 peaks were assigned as the major TSS of annotated protein-coding genes as well as the viral telomerase. Of these, the median focus index (a measure from 0–1 of the narrowness of the peak, with one being a single bp in width) was 0.92, and the lower quartile was 0.83, indicating that the vast majority of the major TSS were composed of sharp, well-defined peaks. The median and lower quartile of peak summed intensities were 21 and 10, respectively. The median 5'UTR length of the 73 genes with major TSS assigned was 85 bp, with a minimum length of 11 (MDV075) and a maximum length of 521 bp (MDV061) (S12 Fig). These calculations take into account spliced 5' UTRs detected in the major transcript isoform of several genes.

The recommended PacBio isoseq3 pipeline, by default, collapses all transcripts that differ only in 5' start site into the longest form, under the assumption that 5'-truncated forms are sequencing artifacts. In herpesviruses, many genes occur in co-terminal clusters, each with a distinct promoter and transcript start site (TSS) but sharing a 3' cleavage and polyadenylation signal [32]. Because of this, these default settings are overly aggressive for herpesviruses, collapsing gene clusters into a single, long transcript. Conversely, without any collapsing, the number of transcript models predicted becomes so large as to be uninformative. After testing several permutations of software settings, we settled on collapsing transcripts with 5' ends within 200 bp of each other. This reduced the noise in the final dataset to a reasonable level while allowing for meaningful quantification of the transcripts in each cluster. However, upon visualization of the resulting data against the alignment of the original FLNC reads, it was clear

that the predicted TSS from many of these intermediate models was not as precise as the data allowed. As an alternative, we tested the Mandalorion tool v4.0.0 [33] for transcript collapsing. On our dataset, Mandalorion produced models with more accurate TSS (based on comparison to FLNC mapping) but completely missed some abundant and well-supported alternative isoforms, even after extensive optimization of the software parameters.

With the ability to assign TSS to roughly three-quarters of the characterized MDV genes at nucleotide-level resolution, we were able to analyze the promoter context of the MDV transcriptome in epithelial skin cells to an extent that has not previously been possible. After predicting the location of core Pol II promoter elements in the reference genome using TFBSTools [34], we categorized each major TSS by the presence or absence of the TATA, initiator (Inr), downstream promoter (DPE), upstream B recognition (BREu), downstream B recognition (BREd), and CCAAT elements in the immediate vicinity, constrained by the known expected spacing of each element to the TSS. Additionally, a second set of 58 secondary TSS, defined as those not assigned to a gene as the major TSS but with a minimum intensity of 5 and a focus index $\geq$ 0.67, were analyzed under the same criteria.

## Core promoter usage in MDV

**TATA box.**    The TATA box is perhaps the earliest known and most well-characterized of the eukaryotic Pol II core promoter elements. When present and functional, it has been previously shown to be constrained to a position between -28 and -34 bp upstream of the TSS, with an optimal spacing of -30 to -31 [35]. We therefore constrained our search to predicted TATA motifs starting between 20 and 40 bp upstream of the putative TSS. Under these criteria, 61 of 73 major TSS (84%) were found to have a TATA motif in the expected location. Of the secondary TSS, 38 of 58 (66%) had a TATA motif in the expected location. We found that TATA spacing in MDV closely follows the rules previously delineated for eukaryotic usage, with the spacing to major TSS tightly distributed between –26 and -34 bp and a median, mean, and mode all of -30 (Fig 8A). The spacing to secondary TSS were slightly more dispersed but still confined between -27 and -37 bp, with a major spike at -31 (Fig 8B). Because TATA motifs were found relatively frequently throughout the genome by TFBSTools at the cutoffs used, we also selected and characterized a set of 180 random positions throughout the genome. Of these, 80 (44%) had an identified TATA motif within 20 to 40 bp upstream. However, the distribution of distances between these random sites and the most optimal upstream TATA box was found to be uniformly distributed (Fig 8C; chi-squared test for uniformity p-value: 0.83; chi-squared p-value of major TSS/TATA spacing: 0.001).

Since the search of the genome for TATA motifs used the JASPAR position-specific weight matrix, some identified sites differed significantly from the consensus motif TATAWAW. The sequence logo for all aligned TATA motifs found upstream of identified TSS peaks is shown in Fig 8D. To test whether there was a relationship between PWM score and deviation from the optimal TATA/TSS spacing, the two variables were plotted against each other for identified TSS as well as randomly selected positions (Fig 8E). We found that, while the putative true TSS sites had a higher overall score distribution than the randomly selected positions, there was not a strong correlation between PWM score and deviation from the optimal -30/-31 spacing. Spacing for randomly selected positions was uniformly distributed in both dimensions. These results suggest that in MDV, spacing between the TSS and TATA motif is more highly conserved than the consensus TATA signal itself.

**Other core promoter elements.**    In addition to the TATA box, we also searched the local context of the identified TSS for the Inr (starting at -2 bp), CCAAT (starting -120 to -50 bp), DPE (starting -34 to -28), BREu (within 3 bp upstream of identified TATA), and BREd (within

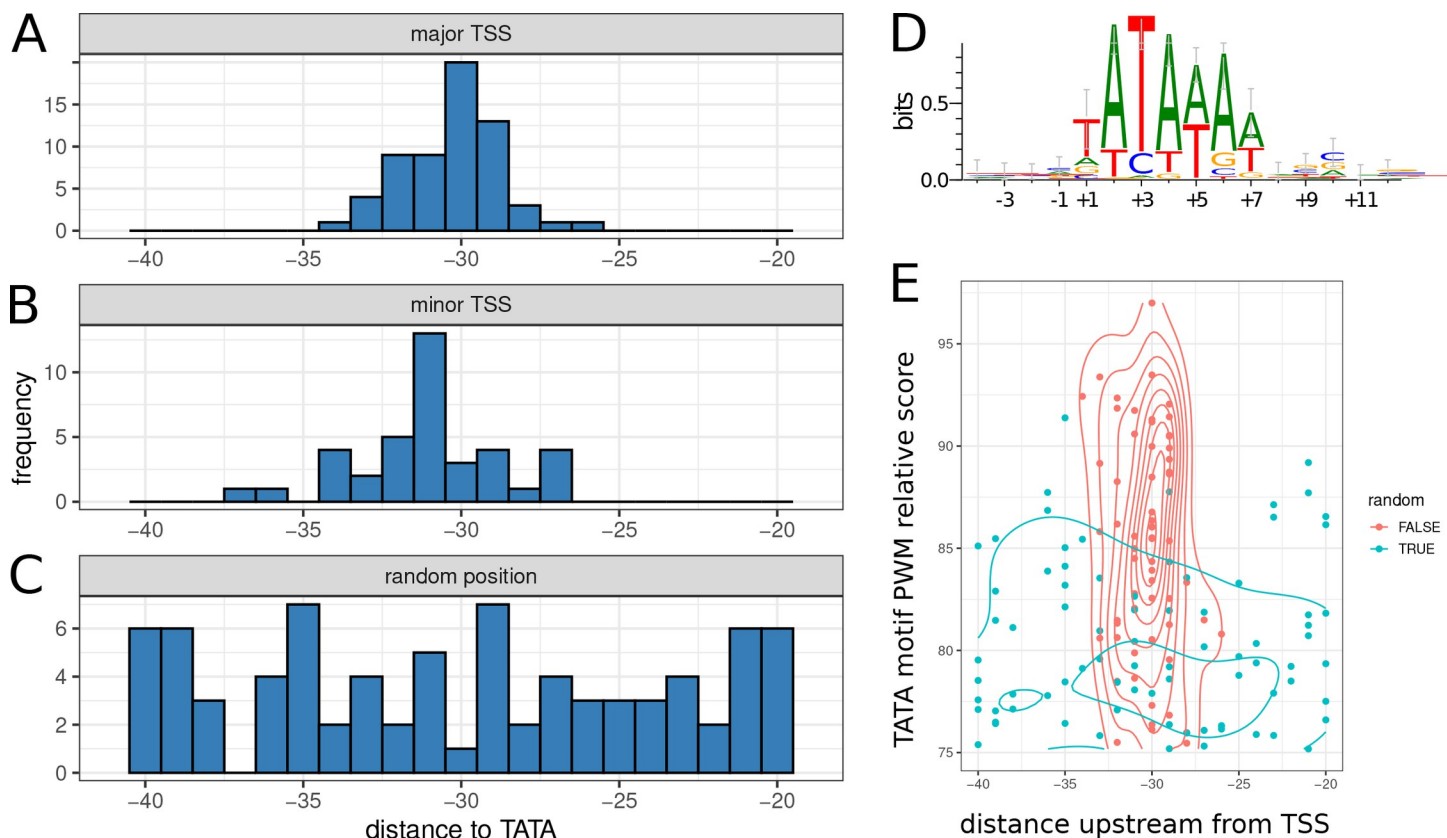

**Fig 8. TATA motif usage in MDV.** (A) Distribution of distances between major TSS and start of optimum upstream TATA motif, with search constrained to positions 20 to 40 bp upstream. (B) Distances between secondary TSS and upstream TATA motif. (C) Distances between randomly selected genomic positions and upstream TATA motif. (D) Sequence logo of aligned identified TATA motifs and surrounding context for all identified TSS peaks. (E) Relationship between TSS/TATA spacing and TATA motif PWM search score. Points from identified TSS peaks are shown in red, and points from randomly selected genomic positions are shown in blue. Lines indicate a 2D kernel density estimation for each class of points.

3 bp downstream of identified TATA). For major TSS, the most frequently found of these motifs was the CCAAT motif, found in 54 of 73 peaks (74%) at the specified search distance. However, this motif was also found under the same restrictions in 122/180 (68%) of randomly selected points.

Similarly, DPE was found downstream of 49 of 73 peaks (67%) at the specified search distance. However, this motif as well was also found in the specified distance range in 74% of randomly selected positions. The BREu and BREd were found in 3 and 8 (4.9% and 13%) of TATA-containing TSS loci, respectively; they were found in 3 and 16 (3.8% and 20%) of 80 TATA-containing randomly selected points. Unlike the well-defined characteristics of TATA box usage, therefore, the similar frequencies of these four core motifs between TSS-specific and randomly selected points calls into question the meaningfulness of their *in silico* predictions in the context of MDV biology in the absence of further functional validation.

The initiator element (Inr) motif, which overlaps the TSS itself and is more specifically defined both in terms of spacing and sequence relative to the TSS, was identified in 20 of 73 primary TSS peaks (27%) as compared to 9 of 180 (5.0%) of randomly selected points. The more general YR (or PyPu) dinucleotide overlapping the TSS (positions -1 to +1) [36] was found in 60 of 73 (82%) of major TSS peaks, and 44 of 58 secondary TSS peaks (76%), compared with 26% of randomly selected points (25% of randomly selected dinucleotides would

be expected to be YR pairs). Sequence logos of the immediate genomic contexts for primary
TSS, secondary TSS, and non-YR-containing TSS are shown in S13 Fig. The YR dinucleotides
are clearly prominent in both primary and secondary putative TSS, while non-YR TSS are
enriched in purines immediately upstream of the putative start site (S13B Fig). Combined
TATA box and YR dinucleotide were found in 51 of 73 (70%) of major TSS and 30/58 (52%)
of secondary TSS, compared with 10% of randomly selected positions.

## Transcript termination and cleavage

Mapping of IsoSeq full-length non-concatemeric reads against the reference genome, in addi-
tion to enabling precise determination of the major transcription start site of the majority of
MDV genes, also enabled precise determination of the 3' transcript end sites (TES) and thus
the sites of pre-mRNA cleavage and polyadenylation. The RefSeq genome annotation for
MDV (isolate Md5; accession NC_002229.3) annotates the majority of MDV genes (60,
excluding duplicate genes in terminal repeats) as occurring in co-terminal clusters of two or
more. The RefSeq record does not indicate how these transcript boundaries were determined
and they were not described in the original publication [16]. In any case, the IsoSeq transcript
build and separate TES calculation in epithelial skin cells confirm nearly all the clusters anno-
tated in the RefSeq record, with several exceptions. The RefSeq annotations include MDV012
in a co-terminal cluster with MDV013 (UL1), MDV014 (UL2), and MDV015 (UL3). While
identifying low levels of MDV012 transcripts extending to a co-terminus with the downstream
cluster in epithelial skin cells, the vast majority of MDV012 transcripts (several orders of mag-
nitude more abundant) terminate 77 bp downstream of the stop codon, prior to the TSS of the
nearest downstream gene (see IGV browser). Similarly, the single transcript model determined
for MDV021 (UL9) terminates 135 bp downstream of the stop codon, rather than sharing a
termination and cleavage signal with the downstream MDV020 (UL8) as in the RefSeq annota-
tion. In both cases there are canonical polyadenylation signals (PAS) located the expected dis-
tance upstream of the determined sites of cleavage (19 and 20 bp, respectively). Conversely,
MDV064 (UL49.5) and MDV062 (UL49) are annotated as sharing a relaxed PAS (ATTAAA)
prior to MDV061 (UL48). In epithelial skin cells, no IsoSeq transcripts were found to match
this model; rather, abundant transcripts for both genes formed a coterminal cluster with the
downstream MDV061 (UL48).

   The recognition signal for pre-mRNA terminal cleavage and subsequent polyadenylation in
eukaryotes has long been understood to consist of a generally well-conserved hexamer motif
(AWTAAA) 10–30 bp upstream of the cleavage site and a less-well-defined downstream U or
G/U rich region within 40 bp of the cleavage site [37,38]. Additionally, the precise site of cleav-
age is frequently immediately downstream of a BA ([C/T/G]A) dinucleotide, where the first
nucleotide preference varies between organisms and kingdoms [39]. Li and Du found that in
chickens, the first nucleotide preference was C>G = T, while in non-mammalian metazoans
overall the preference was T>G = C.

   In order to examine the extent to which MDV transcription in epithelial skin cells followed
these principals, we looked at all TES identified from $\geq 5$ FLNC reads and which were not
flagged as probable genomic priming artifacts by SQANTI3. We identified 111 putative tran-
script cleavage/polyadenylation sites based on these criteria, and of these, 103 (93%) had PAS
hexamers starting within a wide 5–50 bp search window upstream. In fact, actual spacing for
all 103 sites to the upstream PAS fell in the range of 11 to 35 bp, with 96 (93%) occurring in the
range of 15 to 29 bp (S14A Fig). The median, mean, and mode of the PAS spacing were 19, 20,
and 17 bp, respectively. A sequence logo plot of the surrounding sequence context centered on
the putative cleavage sites clearly shows the upstream A/T rich region of the hexamer signal

and the downstream T- (U)-rich motif. The BA consensus motif at the actual site of cleavage is also readily apparent, although it is offset by 1 bp, presumably because the terminal adenosine is trimmed off by the IsoSeq poly-A tail trimming step. In MDV, the preference of the first nucleotide in the dinucleotide motif is roughly T = C>G. Overall, the results confirm that MDV transcript terminal processing follows the established mechanisms and conversely support the accuracy of the determined TES sites in this experiment. The PAS/cleavage spacing is in general agreement with that determined by Bertzbach et al [20], although with different methodology their distribution was shifted towards shorter spacing (range 4–29 bp; median, mean and mode of 14, 14, and 14, respectively).

## Messenger RNA splicing in epithelial skin cells

Novel mRNA splicing has been identified during *in vitro* infection of CECs [19,40] and, more recently, B cells [20], in particular MDV073 (pp38), MDV078 (vCXCL13-vIL8), and MDV027 (UL15-TRM3). As in other transcriptomic studies with MDV-infected cells [19,20], we also detected mRNA splicing events occurring in known coding regions and seemingly intergenic and anti-sense regions (Figs 2–6). Reads spanning some of these introns were detected in our RNA-Seq analysis but at very low abundance, suggesting their expression is limited in epithelial skin cells. Similarly, formerly identified mRNA splicing of MDV076 (Meq) and MDV078.3 (RLORF4) to MDV078 (vCXCL13-vIL8) were detected [41–43], but also at low levels in epithelial skin cells. In contrast to the above splice variants detected at low abundance, pp38A and pp38B transcripts were not detected (Fig 3) in either short read or IsoSeq data, suggesting they are not expressed in epithelial skin cells. In immortalized chicken cells (DF-1), primary chicken cells (CEC and CKC), and splenocytes infected with MDV, the expression of both pp38 and pp38B at the RNA level has been demonstrated, although no proteomic evidence (MS or western blotting) was reported [40]. We identified a novel pp38 splice variant (Novel pp38C) at the RNA level, albeit without evidence from MS/MS discussed below. Only minor splicing events of pp24 were detected in our RNA-Seq data, contrary to what was previously reported by Bertzbach et al. [20]. Overall, it appears increasingly likely that the extent of viral mRNA splicing identified within the IRL/IRS regions may depend on the infected cell type. Viral gene expression is likely more tightly regulated in cells naturally infected by MDV compared to artificial *in vitro* cell culture systems that do not produce infectious cell-free virus.

## Transcriptional regulation via 5' UTR splicing

In addition to confirming known and novel splice events in protein-coding sequences, IsoSeq transcripts revealed splicing in the 5' UTR of several genes which appears to be involved in regulating their expression (see IGV browser). MDV073.4, MDV073 (pp38), and MDV072 all appear to share a common TSS. However, five different detected isoforms (one unspliced, four spliced) appear to produce products where the 5'-most start codon yields either MDV073.4, MDV073, or MDV072. The latter is particularly striking, as removal of a 1748 bp spliced intron completely excises the coding sequences of the upstream genes. The primary transcript of MDV018 (UL6) also has a large intron in the 5' UTR, placing the TSS and promoter region in the middle of MDV017 (UL5) on the opposite strand, and far upstream of where it might be expected to be located. For MDV010 (vLIP), the primary transcript closely matches the known annotated splice form, with only a short 5' UTR of ~ 40 bp. However, a second isoform present at 55% relative abundance shares a TSS with MDV008 (pp24), with the entire MDV008 (pp24) coding sequence spliced out. In MDV037 (UL25), while transcripts representing unspliced products were transcribed from diffuse locations upstream of the start codon, the most abundant transcript shares a TSS with MDV035 (UL24), with the majority of the MDV035 (UL24)

coding sequence spliced out. This situation is slightly more complex, as the start codon of MDV035 (UL24) is included in the 5' exon of this transcript, but out of frame with the downstream MDV037 (UL25) coding sequence. However, the MDV035 (UL24) start codon has a weak Kozak context while the MDV037 (UL25) start has a strong consensus sequence. Taken together, these observations stress the complex nature of transcription control in MDV and suggest an important role for alternative splicing beyond changes in protein structure.

## Confirmation of productive transcript splicing by MS/MS

Decades of gene expression and mRNA splicing studies have identified numerous viral mRNA splicing events during cell culture replication and in MDV-transformed chicken cells, including vIL-8, Meq/vIL-8, pp38, RLORF4, and gC splicing products [25,40,41,44–46]. Although extensive analysis has been performed at the transcriptional level using RT-PCR and sequencing, in addition to traditional protein expression studies using western blotting and immunofluorescence assays, few of these spliced products have been directly validated at the level of peptide identification. Due to the depth of peptide sequencing achieved herein, we could detect high-confidence peptides spanning the previously described intron boundaries of four proteins. These include the second intron of vCXCL13 (RTEIIFALK), UL15 (STVTFASSHNTNSIR), gC104 (DGSLPDHRS[+80]P), and the 14 kDa A nuclear protein (YISYPGCIDCGPTFHLETDTATTR) (Fig 9). Three of these peptide identifications, in addition to q-values < 0.01, are supported by rich b/y ion series and infection-specific elution profiles (S15 Fig). The pUL15 peptide was found only by the Byonic search engine, and the supporting MS2 and XIC plots are poor (S16A Fig). Additional novel splice junctions are described as part of the proteogenomic search results below.

## Validation of translation start sites and assessment of protein N-terminal PTMs

MS/MS database searching of enzymatic digest spectra allows for the identification of protein N-terminal peptides, which lack a cleavage site (in this case, by trypsin/LysC) at the N-terminal end of the peptide. By including the common protein N-terminal PTMs of N-terminal methionine excision (NME) and N-terminal acetylation (NTA) in database searches, we confirmed the translational initiation sites (TIS), as well as PTM state, for 23 viral proteins (S7 Table). Nearly all identified TISs agreed with the existing gene annotations. By including the sequences for potential alternative TISs (based on the start codon and Kozak context) in the search database [47], we could also search for alternative or incorrectly annotated sites. For MDV015 (UL3), we confirmed the use of a downstream start codon as the TIS, in agreement with the RefSeq annotation of Md5 (NC_002229). Although there is an in-frame start codon upstream to this position, it has a weak Kozak context (CAC**ATG**C) compared to the strong Kozak consensus of the true TIS (ATA**ATG**G). For this analysis, a strong Kozak motif was considered to be one matching the consensus sequence (-3)RNN**ATG**G(+4). In addition, we identified a peptide indicating an alternative TIS for MDV055 (UL42), together with a peptide representing the annotated TIS (S17 Fig). The annotated TIS peptide (AGITMGSEHMYDDT TFPTNDPESSWK) was identified by a total of 23 PSMs in all three replicates and in two different charge states (S17B Fig). The alternative TIS peptide (GSEHMYDDTTFPTNDPESS WK) was identified by a total of 12 PSMs in all three replicates and in two different charge states (S17C Fig). Similarly, normalized LFQ intensities for the two peptides were $1.1 \times 10^7 \pm 4.2 \times 10^6$ and $2.4 \times 10^6 \pm 2.5 \times 10^5$ (S8 Table), suggesting the alternative downstream TIS is utilized at roughly one-fourth the rate of the annotated TIS. Both N-terminal peptides underwent NME and NTA. The alternative TIS (S17A Fig) has a slightly stronger Kozak context, with

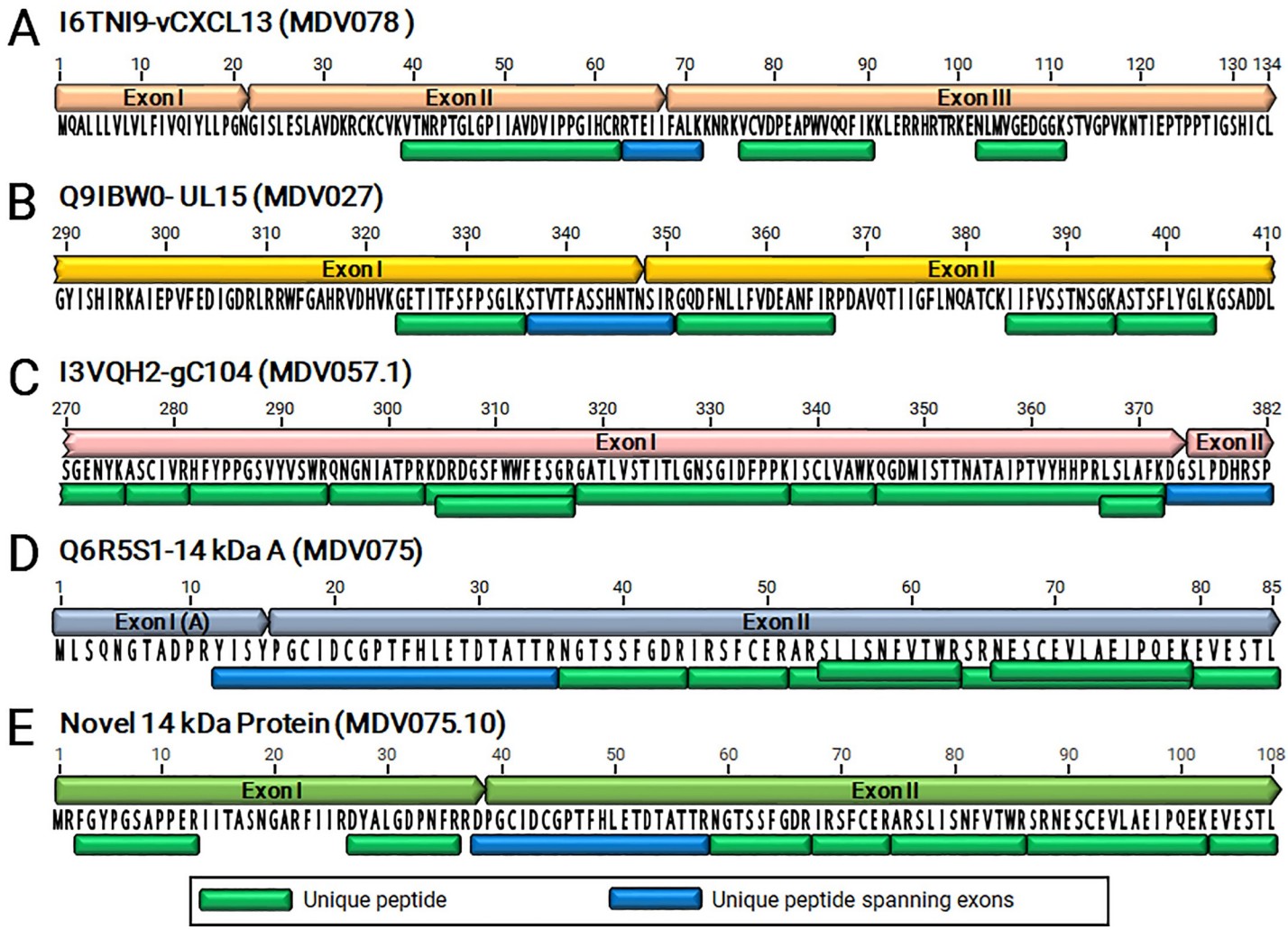

**Fig 9. Evidence of mRNA splice products by unique peptides.** Following MS/MS-based proteomics and analyses, unique peptides spanning exon junctions were identified for vCXCL13-vIL8 (A), pUL15 (B), gC104 (C), 14 kDa A (D), and the newly identified Novel 14 kDa protein (E). Peptides detected using tryptic digestion are shown with peptides spanning exon junctions in dark blue.

conserved bases at both -3 and +4 positions (ACT**ATG**G), while the annotated TIS has a non-conserved -3 base (TCA**ATG**G). Both peptides have strong MS2 ion series and infection-specific extracted ion chromatograms. The biological significance of the alternative TIS of MDV055 (UL42) in epithelial skin cells remains unknown.

Several additional novel N-terminal peptides were detected in the expanded search. A putative alternative TIS peptide ([+42]S[+80]SSTLAQIPNVYQVIDPLAIDTSSTSTK) was found for MDV070 (UL55) in addition to detecting the annotated TIS peptide ([+42]AAGAMS[+80]SSTLAQIPNVYQVIDPLAIDTSSTSTK). In this case, only the alternative TIS (S18A Fig) has a strong Kozak context (GCG**ATG**T). Both peptides have sparse but specific MS2 ion series, and both have extracted ion chromatograms specific to infected replicates (S18BC Fig). The annotated TIS peptide was identified in four PSMs from all three phospho-enriched replicates, while the alternative TIS peptide was identified in two PSMs from two phospho-enriched replicates. Both N-terminal peptides underwent NME and NTA (S7 Table).

Detection of a novel N-terminal peptide (ANINHIDVPAGHSATTTIPR) in MDV096 (US7-gE) would represent translation initiation at a non-canonical start codon in an otherwise strong Kozak context (GGA**ACG**G). The peptide was identified in all three replicates and with a reasonably complete ion series; however, the extracted ion chromatogram shows elution peaks in both infected and mock-infected samples, suggesting that this is likely a misidentified peptide (S16D Fig).

Overall, patterns of NME and NTA followed the expected rules based on the local amino acid context (S7 Table). Of the 23 viral protein N-terminal peptides detected in MS/MS (including alternate starts), 15 were always detected with the N-terminal Met removed. All of these had small +2 amino acids (A, G, S, or T) in accordance with known rules [48]. Six TIS-indicating peptides always retained their N-terminal Met. All of these had larger penultimate residues (D, E, or M). Two termini were identified with peptides both N-terminally cleaved and uncleaved, somewhat surprisingly (M and N +2 residues). Nearly all identified N-terminal peptides were acetylated, either in every PSM (19) or part of the time (3). Only one peptide was detected in an unacetylated state–the N-terminus of MDV044 (UL31), TGHTLVR.

## Quantitative analysis of mRNA splicing and validation at the protein level in epithelial skin cells

The alphaherpesvirus conserved gC, encoded by MDV057 or UL44, has been previously shown to be alternatively spliced to produce two secreted forms called MDV057.1 (gC104) and MDV057.2 (gC145) [25]. The mRNA splicing of UL44 is believed to be conserved, as it has been observed during HSV infection [49]. Importantly, all three MDV gC proteins (gC, gC104, and gC145) are required for efficient horizontal transmission [25], but their expression at the mRNA or protein level had never been examined in epithelial skin cells. All three forms have been detected in RNA-Seq studies in infected cell cultures [19,25] and in B cells infected *in vitro* [20]. Here, we confirmed that both splice variants are expressed in epithelial skin cells based on intron-spanning (spliced) reads and IsoSeq data (Fig 10A). Interestingly, gC104 (MDV057.1) is ~4-fold more abundant than gC145 (MDV057.2) and ~10-fold more abundant than the non-spliced transcript based on intron read depth (Fig 10B), and IsoSeq data confirmed the abundant expression of the splice variants. Five peptides that are unique to the non-spliced product (MDV057) were detected, confirming full-length gC expression (Fig 10A). As discussed above, the unique intron-spanning peptide of gC104 was detected, while no unique peptides were identified for gC145; however, the transcript abundance data suggests it is present, and most peptides would be shared with the other isoforms. Tryptic mapping of gC145 predicted gC145 unique fragments of 3, 4, 1, and 38 aa (Fig 10C). Only the 38 aa peptide would be detectable by MS/MS–it is therefore likely that this proteoform is present but that the single unique peptide was missed. Further studies are needed to determine whether UL44-gC145 is expressed at the protein level (gC145). However, we can confirm gC104 is expressed at both the RNA and protein levels.

The MDV-specific pp24 and pp38 phosphoproteins are encoded by MDV008 and MDV073, respectively, and share identical N-terminal 65 aa sequences (S19 Fig). Additionally, the MDV073 gene has been shown to produce two alternatively spliced mRNA and proteins called spl A and spl B [40], or pp38A and pp38B, respectively, during *in vitro* replication, and they are reported to be important for MD pathogenesis and tumor development [50]. Analysis of intron-spanning reads for this region showed no evidence that pp38A or pp38B were produced in epithelial skin cells; however, a novel splicing product was identified utilizing the pp38A exon I donor splicing site (D1) to a novel acceptor site (A) and spanned by an average of 114±56 reads in the six replicates. This acceptor site was located downstream of

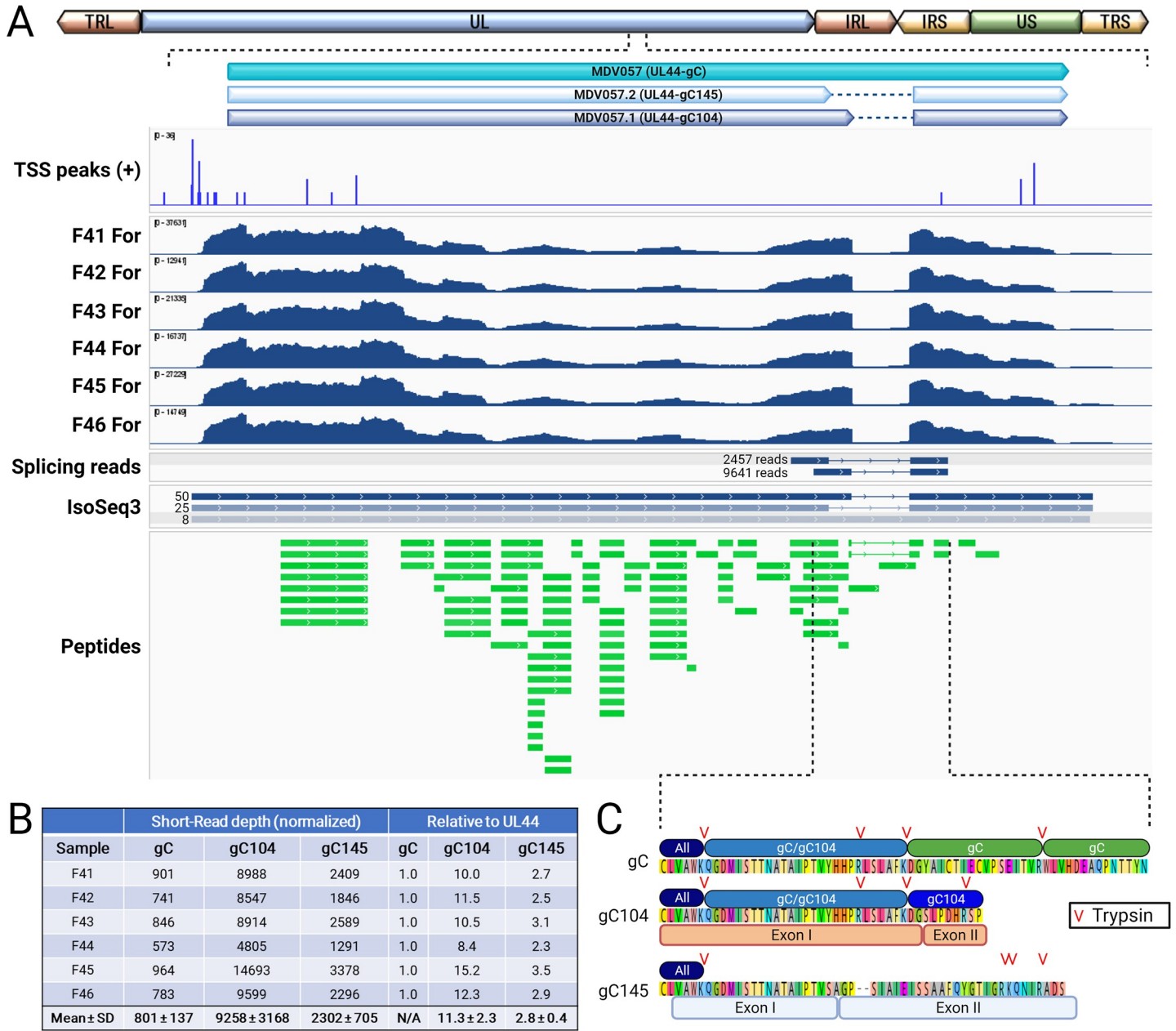

**Fig 10. Quantitative analysis MDV057, MDV057.1, and MDV057.2 mRNA expression and peptide validation.** (A) Schematic representation of the MDV genome with the region encoding MDV057 (gC) expanded from the UL. TSS peaks, read depth of the six infected replicates, intron-spanning (spliced) reads, IsoSeq3 transcripts (combined two infected replicates), and identified peptides. Also shown are read counts for spliced reads and IsoSeq scores for transcripts. (B) Total read counts for intronic reads (MDV057/gC) and intron-spanning reads (MDV057.1/gC104, MDV057.2/gC145) in the six infected replicates and the average reads ± standard deviations are shown in table form. The ratio of MDV0057.1 (gC145) and MDV057.2 (gC145) transcripts compared to MDV057 (gC) ± standard deviations is also shown. (C) The region highlighted in (A) was expanded to show the protein sequences of gC, gC104, and gC145, exon junctions for gC104 and gC145, and unique peptides detected for gC and gC104. Predicted tryptic cleavage sites are also shown for gC104 and gC145. Image created with BioRender.com.

the pp38 ORF stop codon and would code for a protein of 137 aa that we termed Novel pp38C, which only differs from pp38A by the C-terminal 8 aa (S19A Fig).

## RNA expression and protein validation of pUL26.5 in epithelial skin cells

The overlapping ORFs encoding MDV038 (UL26) and MDV039 (UL26.5) pose a differentiation dilemma. The UL26 and UL26.5 proteins are encoded by overlapping transcripts with alternative TIS, with the encoded proteins sharing identical C-termini. Therefore, nearly all the potential peptides of the latter are shared with the former (Fig 11A). UL26 encodes a serine protease (pUL26-SCAF) cleaved during procapsid maturation to yield two proteins, VP24 and VP21, the latter of which is almost identical to pUL26.5-ICP35 [51–56]. Fortuitously, a tryptic peptide of the N-terminus of pUL26.5 was detected in our data with strong elution peaks in infected replicates (Fig 11B). This distinguishable finding, along with the increased RNA-Seq depth (Fig 11A) and more intense LFQ peptide intensities for pUL26.5 vs. pUL26 (S8 Table), strongly suggests pUL26.5-ICP35 is produced from its own TIS in epithelial skin cells. Potential TATA- and CCAAT-boxes are shown in Fig 11C. IsoSeq showed a strong TSS peak (score 31) at position -133 from the start codon (Fig 11C) that is 24 and 90 bp downstream from putative TATA- and CCAAT-box motifs, corresponding to their expected locations [35,57], adding further evidence that UL26.5 is transcribed and translated as its own protein. A similar genetic arrangement with an internal promoter within the body of UL26 has been reported for HSV-1, suggestive of a conserved transcriptional regulatory function [58].

## Evidence for translation of the 132 bp direct repeat reading frame

The hypothetical MDV075.2 ORF is highly variable in length because it spans the 132 bp tandem direct repeats that differ in copy number between strains. The role of the 132 bp repeat region in the pathobiology of MD has been investigated thoroughly since the expansion of this region from 2 copies to over 20 occurs concomitantly with attenuation [59]. It has been reported that this expansion disrupts the 1.8 kb RNA transcript family that contains a putative fes/fps kinase-related transforming protein [60]; however, this expansion was proven insufficient to cause attenuation [61]. It was considered possible that this expansion affected the expression of proteins linked to the 1.8 kb RNA transcript (Fig 7A), but no evidence existed that the 132 bp direct repeats encoded a translated protein.

RNA-Seq results suggest MDV074, MDV075.4, and MDV075.7 at this locus are not expressed abundantly in epithelial skin cells (Fig 4 and S3 Table), and the lack of matching peptides in MS/MS is consistent with this result. However, the region containing the MDV075.2 gene was abundantly expressed (3588 ± 461-fold coverage) in epithelial skin cells (S3 Table). The most likely reason for this depth of coverage is from non-spliced mRNAs of the highly abundant 14 kDa family of transcripts (discussed below) within an intron of which MDV075.2 is situated (Fig 7A). However, two unique peptides from the MDV075.2 protein sequence were identified (FLCLLPQGGGAR and RACS[+80]VTALAR) and provide support for the possibility that this variable-length ORF is translated into a protein product (Fig 7B). Annotated spectra and XIC for the first peptide are shown in S20A Fig. MS1 intensities for the peptide mass are low, but the XIC provides evidence that this ion is specific to infected samples. The matching b/y ions are moderate, although the strong $y_7$ ion peak corresponding to an N-terminal proline agrees with the expected pattern from the known "proline effect" [62]. The XIC of the second peptide shows strong peaks in both infected and mock-infected samples; this is likely a mis-assigned PSM (S20B Fig). Nevertheless, the data suggests the possibility that MDV075.2 within the 132 bp direct repeat is expressed, and the implications of the expansion of this region during attenuation are intriguing.

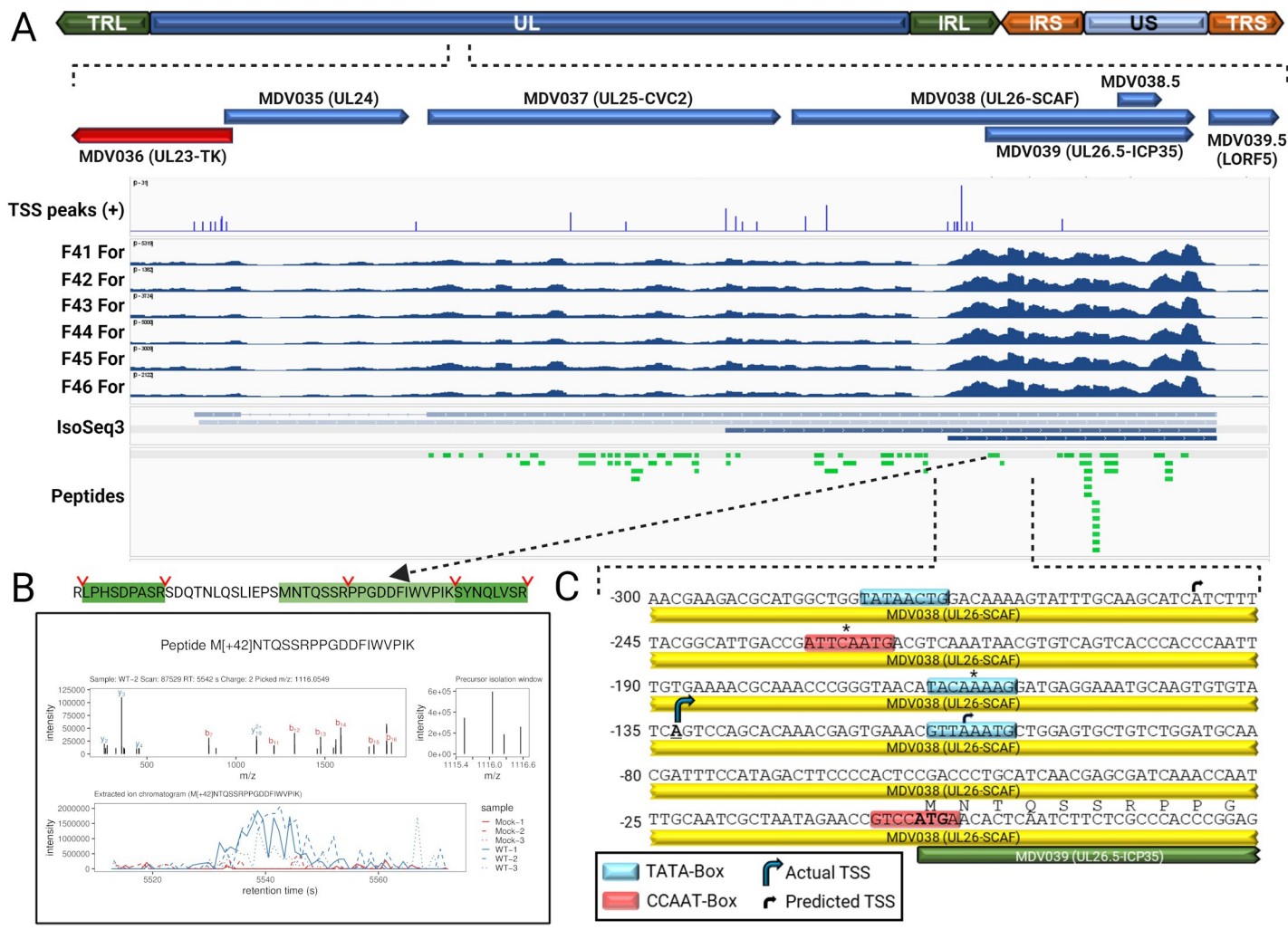

**Fig 11. Expression of pUL26.5 in epithelial skin cells.** (A) Schematic representation of the MDV genome with the region spanning MDV036 (UL23) to MDV039.5 (LORF5) expanded from the UL. TSS peaks, read depth of the six infected replicates, IsoSeq3 transcripts (combined two infected replicates), and peptides identified are shown for the forward strand. (B) The N-terminal peptide of pUL26.5 detected in MS/MS with tryptic cleavage sites is shown, with the annotated MS2 scan and extracted ion chromatograms from the six replicates. (C) Predicted TSS and TATA- and CCAAT-boxes. The actual TSS was identified based on IsoSeq TSS peaks that would predict actual TATA- and CCAAT-boxes denoted with an asterisk (*). Image created with BioRender.com.

## Alternative splicing in the 14 kDa family of transcripts

The repeat long region of MDV has a complex arrangement of genes and expression patterns, including miRNAs [63,64], internal ribosomal entry sites (IRES) [65], and circRNAs [66]. Of importance in the context of this report is the examination of mRNA splicing in the region spanning the 14 kDa family of nuclear proteins (Figs 4 and 7A), whose transcripts are amongst the most highly expressed in epithelial skin cells (S3 Table). We detected abundant intron-spanning reads for both the known 14 kDa A and 14kDa B splice forms, in addition to a third splice variant with a novel exon I (Fig 7A). Exon I of this novel isoform is comprised of the 5' half of the hypothetical ORF previously annotated as MDV075.6 (Figs 4 and 7B). In epithelial skin cells, isoform A (MDV075) is the predominant RNA species, while the novel isoform is half as abundant, and isoform B is one-fourth as abundant as isoform A (Fig 7A). IsoSeq read counts were consistent with this observation.

Previous studies have shown that 14 kDa A and B were expressed at the protein level using polyclonal antibodies generated against each protein, but these studies could not differentiate 14 kDa A and B due to their shared protein sequences and lack of distinguishing peptide differences (39). We identified the intron-spanning peptide of the 14 kDa A isoform in the MS/MS dataset (Figs 7C and 9) but not of the B isoform. Importantly, the proteogenomic scan identified the peptide spanning the splice junction of the novel 14 kDa isoform (DPGCIDCGPTFH LETDTATTR) shown in Figs 7C and 9, putatively termed MDV075.10. This peptide is supported by rich MS2 spectra and infection-specific extracted ion chromatogram profiles (S21 Fig). It should be noted that the same peptide sequence is found spanning the B isoform splice junction, but in that protein it is not tryptic, as there is no Arg (R) or Lys (K) immediately upstream, and thus it would not be detected in this assay (Fig 7C). Based on LFQ intensity, the A isoform-spanning peptide is ~5× more abundant than the novel spanning peptide ($4.5\times10^6$ ± $9.1\times10^5$ vs. $8.9\times10^5$ ± $2.6\times10^5$) (S8 Table). Both intron-spanning peptides were detected in all three infected replicates; neither was detected in any mock-infected samples. Two additional peptides from all three infected replicates were identified from the previously annotated MDV075.6 ORF, now forming the 5' exon of this novel splice product (Fig 7C). Thus, the repertoire of the 14 kDa family of transcripts appears to be even more complex than previously thought, and its expression in epithelial skins cells should be further investigated.

## Novel peptides identified by proteogenomic search

In addition to the novel translation start sites and spliced protein isoforms discovered by proteogenomic searching, several peptides were matched to the six-frame translation of the genome in previously unannotated ORFs (S9 Table). All these spectra passed the 1% PSM FDR threshold, but as with all novel peptides discussed herein, they require extra scrutiny. Of these, most were matched to single PSMs/replicates and have poor or inconclusive MS2 ion series and elution profiles. Two more (IS[+80]LNIR and TGN[+1]NISNNR) have strong elution peaks in both infected and mock-infected replicates and are clearly misidentifications (S16BC Fig). However, the remaining three novel peptides are of interest.

The peptide EEFYEIYFEGCGSRSPTAR has an infection-specific XIC but very sparse MS2 spectra (S22 Fig). However, it matches to the short protein sequence of the recently described SORF6 expressed transcript [20]. As in that publication, we also observed the splice junction associated with the SORF6 transcript in the RNA-Seq data (153 ± 30 spanning reads) and poly-A tailed reads mapping to the putative cleavage site downstream of a canonical polyadenylation signal. Considering a recent publication [30] in which the authors could not detect the translation of a tagged SORF6 coding sequence, the peptide evidence reported here, although not conclusive, beckons further exploration of the protein-coding potential of this ORF.

The two remaining novel peptides mapped to the 5' ends of core genes on the same strand but out of frame. The peptide LEVDHAIVYR maps near the start of MDV060 (pUL47) within a short ORF having a start codon slightly upstream of the core gene (S23 Fig). There are two start codons upstream of the identified peptide, the furthest with a moderate Kozak consensus (AGT**ATG**C) and the second with a strong consensus (GGT**ATG**G). Starting from the upstream codon, a protein of 72 amino acids is predicted, with no known conserved functional motifs. This peptide has both a complete y-ion series and infection-specific XIC elution profiles (S23B Fig). Similarly, peptide FPAAPS[+80]PLPIAHAPVGLDSTR matches a small ORF overlapping the 5' end of ICP4 (MDV084) (S24A Fig). This ORF has a *very* strong Kozak consensus (ACC**ATG**G) and codes for a putative 137 amino acid polypeptide with no known homology or functional motifs. This peptide also has reasonably strong support from the inspection of MS2 spectra and XIC (S24B Fig).

## Proteins notably missing in MS/MS

There were 84 proteins with at least one unique peptide detected. Of these, 2 are hypothetical proteins (MDV075.2 and MDV091.5). Our MS/MS analysis failed to detect peptides for 51 hypothetical and 17 annotated proteins (S3 Table). Of the ORFs with read coverage significantly above background but without peptides detected in MS/MS, several are of note. MDV015.5 (V57) lies at the 3' end of the co-terminal transcript cluster containing MDV013 (UL1-gL), MDV014 (UL2), and MDV015 (UL3), all of which had high peptide coverage in this experiment (Fig 2). However, there are only two predicted tryptic peptides ≥ 6 aa in the UL15.5 protein sequence, making it reasonable to suppose that it was not detected by MS/MS due to the platform's technical limitations. Similarly, MDV072.5 (UL56) lies in a transcript cluster between MDV073 (pp38), MDV072 (LORF11), and MDV071 (LORF10), all of which have moderate to high peptide coverage (Fig 3). This protein has only three predicted tryptic peptides ≥ 6 aa, from 27 to 58 aa long, again making it possible that it was missed due to technical limitations.

MDV056 (pUL43) also lies between two genes with high peptide coverage (pUL42 and pUL44-gC). In this case, unlike the short proteins above, it is predicted to generate 16 tryptic peptides (S25 Fig). However, it is a membrane protein of which 48% of the residues are predicted to lie within 11 transmembrane domains. It should be noted that Liu et al. [23] detected one peptide (MDSVNNSSLPPSYTTTGR) at the N-terminus of the protein in their study in cell culture. The overall hydrophobic nature of the protein may make it unamenable to detection with the methods used here. Alternatively, while the primary IsoSeq transcripts for the flanking genes UL42 and UL44 are very abundant (428 and ~ 90 reads, respectively), the primary UL43 transcript is only covered by 15 reads. A similar situation was observed for MDV032 (pUL20) (S26 Fig), although, unlike MDV056 (pUL43), the read depth for that gene in this experiment was barely above the baseline threshold (S3 Table). MDV075.8, located at the 3' end of the 14 kDa protein family transcript, was another gene with high RNA-Seq levels but no peptide coverage. There are eight predicted detectable tryptic peptides in the protein sequence, and it seems likely that it would have been detected if present at significant levels in the samples. Of note, this ORF contains a weak Kozak consensus (TGC**ATG**T), with conserved residues in neither the -3 nor +4 positions. The two remaining ORFs with high mRNA expression but no peptide coverage, MDV083 and MDV086, lie within the latency-associated transcript (LAT). Here, there is strong evidence of transcription in epithelial skin but no evidence of translation, in agreement with the accepted role of LAT as a non-coding transcript involved in transcriptional regulation [67].

Another protein not detected in our study was MDV035 (pUL24), upstream of the well-represented pUL25 (52% peptide coverage). Liu et al. [23] detected four unique peptides from this protein during cell culture replication, while Bertzbach et al. [20] did not. There are 22 predicted observable tryptic peptides in the protein (S27 Fig), and it is somewhat surprising that none were detected in epithelial skin cells despite relatively low mRNA levels (217 ± 33 fold-coverage). In other herpesvirus proteomics studies, this protein has also been difficult to detect [20,23,27,28,68]. As a rule, some proteins and peptides are inherently more difficult to detect using shotgun LC-MS/MS. Bell et al. failed to detect HSV-1 proteins UL11, UL20, UL43, and UL49.5 [27]. Loret *et al.* [69] failed to detect HSV proteins UL20, UL43, and UL49.5 (gN) in extracellular virions using shotgun proteomics but using the more sensitive targeted multiple reaction monitoring technique were able to detect pUL20.

## Viral genes notably not expressed in epithelial skin cells using short-read RNA-Seq

There were 38 "annotated" ORFs in this study with read depths below the threshold used as a measure of expression (S3 Table). Of these, 30 are annotated to encode "hypothetical proteins" that are nearly all < 200 aa in length (S28 Fig). Whether bona fide functional transcripts or not, our data suggest that they are most likely not expressed in epithelial skin cells. It is possible some of these genes, particularly ORFs within the repeat regions, may be more robustly expressed in lymphocytes. However, Bertzbach *et al.* [20] also found many of these genes to be "not expressed" using an *in vitro* B cell infection model.

Of the remaining annotated ORFs with read depths below the threshold that have been characterized or at least described previously, the most notable are the two helicase-primase subunits, MDV017 (UL5) and MDV066 (UL52). Neither transcript is present above background levels, but both are clearly expressed in epithelial skin cells based on the multiple peptides detected in MS/MS (S2 and S3 Tables). Their protein abundance is well below that of their neighboring genes, and it is likely that they are being expressed at low levels which would be more readily detectable under a different experimental approach. A matching transcript was identified using IsoSeq for MDV017 (UL5) but not MDV066 UL52). Similarly, MDV072 (LORF11) was detected by multiple MS/MS peptides and is also likely to be expressed in skin cells at low levels. This gene was identified in the IsoSeq data and its 5' UTR splicing is discussed in a previous section. Of the remaining genes, MDV013.5 (LORF4), MDV039.5 (LORF5), MDV049.5 (LORF6), MDV050.5 (LORF7), MDV071.4, MDV072.2, MDV072.4, MDV072.6, MDV075.91, MDV080.5, MDV086.2, MDV086.4, MDV097.3, and MDV097.6, there was no evidence of transcription or translation in all studies thus far (S10 Table), and they are unlikely to be expressed during MDV replication. Notably, most are antisense to known expressed genes in the surrounding genomic context.

## Viral genes notably not detected in long-read RNA-Seq

Nearly all well-characterized viral genes were represented by at least one IsoSeq transcript (Figs 2–6). Because longer cDNAs are more difficult to sequence with the IsoSeq library protocol used [70], and with a real-world upper length limit of 7–8 kb [71], some genes with long predicted transcripts which were detected in short-read sequencing and/or MS/MS were not represented, possibly due to these length limitations. These genes include MDV020 (estimated transcript length ~ 5 kb), MDV029 (~ 8 kb), MDV031 (~ 5 kb), MDV049 (~ 10 kb), MDV050 (~ 13 kb), MDV066 (~ 6 kb), and MDV084 (~ 8 kb). Some of these transcripts had one or several overlapping FLNC reads, but the total count was below the final filtering threshold used for reporting and visualization. Some were near or below the threshold for detection in short-read sequencing as well (MDV020, MDV029, MDV050, MDV066). The short MDV065/UL51 transcript, which was reasonably abundant in short-read sequencing and had high peptide coverage in MS/MS, did not show up in the final transcript build. It did have four FLNC reads covering it with shared 3' ends, but the 5' start locations were disparate enough that a passing representative transcript was not reported.

## Conclusions

This study provides a comprehensive analysis of the transcriptional and translational profile during replication of a skin-tropic herpesvirus in the host. To our knowledge, this is the first study in which both short- and long-read RNA-Seq and MS-based proteomics were employed in a natural herpesviral host model supporting the production of infectious cell-free virus

required for host-to-host transmission. While we detected 114 viral ORFs with read depths above our threshold for expression, many of these are likely untranslated ORFs residing within transcribed loci. The 84 proteins (one or more peptides) or 79 proteins (2+ distinct peptides) detected with MS/MS are, therefore, likely a better representation of the protein-coding transcriptional landscape during MDV replication compared to *in vitro* studies (S10 Table).

We have demonstrated herein the application of a method to isolate virus-rich epithelial skin cell samples to maximize virus/host ratios and deeply interrogate the proteome and transcriptome of productive infection. The demonstrated ability to reproducibly detect and quantify nearly all of the viral proteins expressed at this critical stage of infection, as well as obtain a high breadth of peptide coverage for many of them, opens up new possibilities to study viral protein functions in the natural host whereby the effects of mutagenesis/perturbation at the protein and post-translational level can be directly measured. The viral enrichment strategy may further complement additional approaches, such as direct modified RNA sequencing and data-independent-acquisition MS/MS, to continue to push beyond simple gene expression and examine the finer aspects of virus/host molecular interactions.

## Materials and methods

### Ethics statement

All animal work was conducted according to national regulations. The animal care facilities and programs of UIUC meet all the requirements of the law (89–544, 91–579, 94–276) and NIH regulations on laboratory animals, comply with the Animal Welfare Act, PL 279, and are accredited by the Association for Assessment and Accreditation of Laboratory Animal Care (AAALAC). All experimental procedures were conducted in compliance with approved Institutional Animal Care and Use Committee protocols. Water and food were provided *ad libitum*.

### Recombinant (r)MDV

For short-read and MS data, vCHPKwt/10HA was used in which a 3×Flag and 2×HA epitopes were inserted in-frame of MDV UL13 (CHPK) and US10 at their C-termini, respectively, in addition to expressing pUL47eGFP [72]. For long-read sequencing, data obtained from a separate project was used in which UL54 (pICP27) was epitope tagged with 3xFlag at the N-termini and expressed RLORF4mRFP, termed v3×Flag54 [73]. Both viruses are based on the RB-1B strain of MDV [74].

### Animal experiments

For both experiments, Pure Columbian (PC) chickens were obtained from the UIUC Poultry Farm (Urbana, IL) and were from MD-vaccinated parents (Mab+). Fifteen chicks were infected at three days of age with 2,000 PFU of cell-associated virus by intra-abdominal inoculation. Another fifteen age-matched, mock-infected chicks were housed in a separate room.

To monitor the relative level of MDV in the chickens' feathers during the infection, two flight feathers were plucked from each wing (4 total) starting at 14 days pi, fixed in 4% paraformaldehyde for 15 min, then washed twice with phosphate-buffered saline (PBS). Expression of pUL47eGFP (experiment 1) or RLORF4mRFP (experiment 2) were examined as previously described [73,75,76] using a Leica M205 FCA fluorescent stereomicroscope with a Leica DFC7000T digital color microscope camera (Leica Microsystems, Inc., Buffalo Grove, IL, USA). Chickens with heavily fluoresced feathers, along with age-matched mock-infected birds, were euthanized to collect wing feathers for RNA and protein extractions [77]. All samples were collected between 21–35 days pi. Six chickens (replicates) of infected and mock-infected

groups were used for short-read RNA sequencing, while three replicates of each group were used for LC/MS-MS in experiment 1. In experiment 2, two chickens (replicates) and one mock-infected chicken were used for long-read RNA sequencing.

## RNA extraction

The calamus of the feather tips collected were clipped with sterile scissors and dropped directly into 3.0 ml of RNA STAT-60 (Tel-Test, Inc., Friendswood, TX, USA), snap-frozen on dry ice, and stored at -80˚C until all samples were collected. Samples were thawed at 37˚C, mixed with a handheld homogenizer, and 1.0 ml transferred to Phasemaker Tubes (Invitrogen, Waltham, MA, USA) containing 200 μl of chloroform. The samples were vigorously mixed, incubated at room temperature for 3 min, then centrifuged (12,000 x $g$ for 15 min at 4˚C). Total RNA was precipitated with 500 μl isopropanol, washed with 75% ethanol, and dissolved in RNase-free water. The RNA quantity was determined using a Qubit RNA High Sensitivity Assay kit (Thermo Fisher, Suwanee, GA, USA), and its quality was determined using a Bioanalyzer 2100 (Agilent, Santa Clara, CA, USA).

## Short-read RNA sequencing

High-quality RNA samples with RIN values >7.0 were depleted of rRNAs using QIAseq FastSelect–rRNA HMR kit (Qiagen, Germantown, MD, USA) in combination with the KAPA stranded mRNA seq kit (Kapa Biosystems, Wilmington, MA, USA). Ribo-depleted RNA was suspended in the Fragment/Prime/Elute mix and fragmented at 94˚C for eight min. Using the same KAPA kit, cDNAs were generated using random hexamer priming, end-repaired, and indexed with individual adaptors. Libraries were quantified using a Qubit fluorometer and analyzed on a Bioanalyzer 2100 to determine the size distribution of the library. Pooling cDNAs with fragments (200–300 bp) was done using qPCR concentrations. The quality of the final pool was determined using Qubit, fragment analyzer, and qPCR. RNA libraries were prepared for sequencing on Illumina NextSeq 500 instrument using Illumina's dilute and denature protocol. Pooled libraries were diluted to 2nM, then denatured using NaOH. The denatured libraries were further diluted to 2.2pM, and PhiX was added to 1% of the library volume. Data were demultiplexed and trimmed of adapter sequences.

## Short-read RNA-seq data analysis

**Preprocessing and read mapping.** Raw paired RNA-Seq reads were preprocessed using Trim Galore v. 0.6.6 (https://www.bioinformatics.babraham.ac.uk/projects/trim_galore/) in two-color mode, minimum quality 8, minimum trimmed length 40, automatic adapter detection. Reads were mapped against the combined host (bGalGal1.mat.broiler.GRCg7b) and RB-1B (modified from MT272733 based on epitope tags incorporated) genomes using the splice-aware mapper HISAT2 v. 2.2.1 [78], maximum intron length = 50000, strandedness = RF. The RB-1B reference used throughout was trimmed to remove redundant terminal repeat sequences, except for short regions surrounding the TRL/UL and US/TRS junctions which were included to retain putative junction-spanning genes. The resulting alignment files were filtered for reads mapping to the viral genome and split into four strand-specific subsets (forward read/forward strand, forward read/reverse strand, reverse read/forward strand, and reverse read/reverse strand) using SAMtools v. 1.14 [79], filtering on the SAM flags 0×10, 0×40, and 0×80. For all further steps, only the six infected replicates were used (read counts from the mock-infected replicates mapping to the MDV genome were zero or near-zero–see S4 Table). Read-spanning intron junctions were extracted from the BAM alignments using the

RegTools [80] "junctions extract" command (-a 15 -m 20 -M 50000 -s 1). Only junctions supported by ten or more reads per replicate were used for further analysis and visualization.

**Calculation of library strand specificity.** Reads were mapped against the combined host and viral genomes using HISAT2 as above, only no strand specificity was given. Percentage of reads mapping to annotated exons in the sense and antisense orientations was calculated using the infer_experiment.py script from RseqC v. 4.0.0 [81]. For calculation of host transcript strand specificity, exons from the bGalGal1.mat.broiler.GRCg7b reference annotation on chromosome 1 were used as input to the software. For calculation of viral transcript strand specificity, high-quality transcript models as calculated from analysis of the long-read IsoSeq data and described elsewhere herein were used. RseqC was instructed to use all mapped reads for calculation (no subsampling).

**Gene read coverage and background calculation.** Because a high degree of intergenic and/or non-strand-specific mapping to the viral genome was detected in preliminary analysis (see results for further details), it was decided to use background-subtracted median read depth per gene as a metric for evaluating gene transcriptional status. To this end, strand-specific per-base read depth in bedgraph format was calculated using the BEDTools v. 2.30.0 tool genomecov [82]. Strand-specific per-sample median read depths for each annotated MDV gene were then calculated using the BEDTools map command. Inter-sample normalization of gene read depths was performed using median centering, after which overall means and standard deviations were calculated for each gene. Calculation of the background/non-specific read depth was performed as follows. First, because the untranslated regions of the MDV gene models are not well-defined, gene intervals were estimated by adding 600 bp upstream and 100 bp downstream of the annotated coding sequence coordinates using the BEDTools "slop" command. This file was used to mask the full genome coverage bedgraph file using the BEDTools "subtract" command, resulting in an array of per-base intergenic/antisense read depths. Because the distribution of these values was assumed to be a mixture of at least two groups (a large group of true intergenic/antisense positions and a smaller group of actually transcribed positions), the "normalmixEM" method from the R mixtools package v. 1.2.0 [83] was used to generate a preliminary parameter estimate of the true intergenic read depth distribution, assuming a Gaussian distribution. These estimates were further adjusted manually by visualization in R to the final values of mean 50×, standard deviation 33× (S6 Fig). A threshold read depth of mean plus two standard deviations (116×) was used subsequently to determine genes which were expressed above background levels with ~98% confidence (single-tailed).

**Long-read IsoSeq library preparation and sequencing.** RNA was isolated and quantified as for short read sequencing above. Samples with RIN ≥ 7 were used to prepare the IsoSeq barcoded libraries following the PacBio protocol (Procedure & Checklist–Iso-SeqExpress Template Preparation for Sequel and Sequel II Systems, PN 101-763-800 Version 02) using the SMRTbell Express Template Prep kit 2.0 (PacBio). The individual libraries were barcoded using the Iso-Seq Express Oligo kit (PacBio PN 101-737-500). The barcoded libraries were multiplexed in equimolar amounts into a final SMRTbell template, which was purified using 1X beads. The final library was sequenced on the Sequel II system for 20 hours using the Sequel II Binding kit 2.1. The final loading concentration was 80 pM.

## IsoSeq data analysis

**Data preprocessing and transcript calling.** Raw PacBio subreads were processed to circular consensus sequencing (CCS) reads using *ccs* v6.4.0 (--min-rq 0.9) (this and all other PacBio software referenced was installed from the PacBio Bioconda packages https://github.com/PacificBiosciences/pbbioconda) [84]. Demultiplexing and adapter removal was performed

using *lima* v2.7.1 (--isoseq–peekguess). Poly-A tail removal and concatemer filtering were performed using *isoseq refine* from the isoseq3 package v3.8.2 (--require-polya). This step and subsequent transcript clustering were performed using the combined replicate data as recommended by PacBio and replicate read counts were extracted later. Full-length, non-concatemeric (FLNC) reads were then clustered into nonredundant transcripts *using isoseq cluster* (--use-qvs). Transcript cluster sequences were mapped against the combined chicken/RB-1B genome reference (described earlier) using *pbmm2 align* v1.10.0 (--preset ISOSEQ–sort). Overlapping, exon-sharing transcripts were then further collapsed using *isoseq3 collapse* (--do-not-collapse-extra-5exons–max-5p-diff 200) (see Discussion for an explanation of why these settings were used).

**Transcript filtering.** For further analysis, only transcripts with a minimum support of four FLNC reads combined between all replicates were considered. Further filtering of potential artifacts was performed using SQANTI3 v5.1.1 [85]. Initially, filters were applied to remove transcripts with a genomic adenine content downstream of the mapped TTS of 60% or greater (suggesting artifactual priming on the genome rather than a transcript poly-A tail) and spliced transcripts with non-canonical donor/acceptor sites. Later, it was determined that two probable transcripts (based on presence of a canonical poly-A signal at the expected distance upstream of existing genome annotations) terminated at A-rich genomic locations which failed this filter, and specific exceptions were added to the SQANTI3 filter for these two sites.

**132-bp repeat quantification.** The BEDTools "intersect" command (-wa -split) was used to extract FLNC reads from each replicate overlapping the 132 bp repeat region of the reference genome. The CIGAR string of each overlapping read was parsed to extract all insertions (I) and deletions (N) > 130bp, and any indels within 1 bp of an exact multiple of 132 were tallied (*e.g.*, an insertion of 264 bp was tallied as +2, a deletion of 131 bp was tallied as -1). The histogram of these values was visualized using ggplot2 [86] in R.

**Calculation of precise transcript start and end sites.** The BAM mapping file of FLNC reads to the reference genome described earlier was used to calculate frequencies of mapped read starts and ends at each genomic position, after filtering secondary mapped reads and those containing hard or soft clipping at the 5' end (for TSS calculation) or 3' end (for TES calculation) based on CIGAR string. TSS peaks were then called using the FocusTSS algorithm as described previously [87], slightly modified. Briefly, after sorting all genomic positions by decreasing read start frequency, each position was considered as a possible TSS as follows. Frequencies at all adjacent positions within 3 bp were determined, and the position was only considered if it had the highest frequency amongst these adjacent positions, or, in the case of a tie, if it was the 5'-most position. Next, the position frequency was compared to the frequencies of all positions within 50 bp upstream, and it was only retained if the ratio between the candidate frequency and the highest upstream frequency was > 0.1. Then, for the candidate peak, a combined intensity was calculated as the sum of frequencies of all positions within 3 bp, and only peaks with a summed intensity (read start count) of at least 5 were retained. Finally, a "focus index" between 0 and 1 was calculated for the peak as the ratio between the combined intensity and the sum of all position frequencies within a wider window +/- 12 bp of the candidate position. This index was used as a filter for some downstream analyses as described in those sections, in order to select only narrow (highest confidence) peaks. Transcript end site (TES) peaks were localized to all positions where a minimum of 5 FLNC reads terminated, after filtering for intragenomic priming.

**Calculation of promoter signals.** For characterization of the genomic context of putative TSS, the locations of core Pol II promoter motifs in the genome were determined using the TFBSTools R package v1.36.0 [34]. The JASPAR 2020 database was used, and the POLII collection of core promoter position weight matrices (https://jaspar2020.genereg.net/collection/

POLII/) was searched against the RB-1B reference with a minimum score of 75%. Identified motifs were exported from R to GFF3 format for further use.

**Protein extraction, proteomics, and phosphopeptide analyses.** Feathers from MDV-infected and age-matched mock-infected birds were plucked, placed into ice-cold PBS, and epithelial skin scrapings were provided to the University of Illinois Protein Sciences Facility as frozen samples. They were subsequently lysed in a buffer containing 6 M guanidine HCl, ten mM tris(2-carboxyethyl)phosphine HCL, 40 mM 2-chloroacetamide, and 0.1% sodium deoxycholate and then boiled to promote reduction and alkylation of disulfide bonds, as previously described [88]. The samples were cleared of debris by centrifugation and subjected to chloroform-methanol precipitation to remove lipids and other impurities; the resulting protein pellets were dissolved in 100 mM triethylammonium bicarbonate with sonication. Protein amounts were determined by BCA assay (Pierce, Rockford, IL) before sequential proteolytic digestion by LysC (1:100 w/w enzyme: substrate; Wako Chemicals, Richmond, VA) for 4 h at 30˚C and trypsin (1:50 w/w; Pierce) overnight at 37˚C. Peptide samples were desalted using Sep-Pak C18 columns (Waters, Milford, MA) and dried in a vacuum centrifuge. For phosphorylation analysis, phosphopeptides were enriched by iron-immobilized metal ion affinity chromatography (Fe-IMAC) in a microtip format before being desalted once more using StageTips [89].

Peptide digests were analyzed using a Thermo UltiMate 3000 UHPLC system coupled to a high resolution Thermo Q Exactive HF-X mass spectrometer. Peptides were separated by reversed-phase chromatography using a 25 cm Acclaim PepMap 100 C18 column maintained at 50˚C with mobile phases of 0.1% formic acid (A) and 0.1% formic acid in 80% acetonitrile (B). A two-step linear gradient from 5% B to 35% B over the course of 110 min and 35% B to 50% B over 10 min was employed for peptide separation, followed by additional steps for column washing and equilibration. The MS was operated in a data-dependent manner in which precursor scans from 350 to 1500 m/z (120,000 resolution) were followed by higher-energy collisional dissociation (HCD) of the 15 most abundant ions. MS2 scans were acquired at a resolution of 15,000 with a precursor isolation window of 1.2 m/z and a dynamic exclusion window of 60 s.

The raw LC-MS/MS data was analyzed against the Uniprot GaHV2 database (taxon 10390; 1300 sequences) using the Byonic peptide search algorithm (Protein Metrics) integrated into Proteome Discoverer 2.4 (Thermo Scientific). Optimal main search settings were initially determined with Byonic Preview (Protein Metrics) and included a peptide precursor mass tolerance of 8 ppm with fragment mass tolerance of 20 ppm. Tryptic digestion was specified with a maximum of 2 missed cleavages. Variable modifications included oxidation/dioxidation of methionine, acetylation of protein N-termini, deamidation of asparagine, conversion of peptide N-terminal glutamic acid/glutamine to pyroglutamate, and phosphorylation of serine, threonine, and tyrosine. A static modification to account for cysteine carbamidomethylation was also added to the search. PSM false discovery rates were estimated by Byonic using a target/decoy approach.

## Additional proteogenomic analysis

To search for potential novel expressed reading frames and proteoforms, three additional MS/MS search databases were generated from the RB-1B genome sequence using in-house software. A database of tryptic peptides spanning all putative transcript splice sites identified from RNA-Seq was generated, adding an ambiguous residue (X) at each end to prevent the search engine from assuming a protein terminus. A second database containing a full six-frame translation of the genome, split at stop codons, was generated, again adding an ambiguous base at

the N-terminus to prevent identification as a protein terminus. A third database was generated containing possible alternative N-terminal peptides based on potential alternative translation initiation sites (TIS) as follows. For each annotated gene model, all in-frame moderate Kozak consensus sequences (A|G at -3 position, ATG|CTG|GTG|ACG|ATA|TTG|ATT at +1–3, G at +4) were identified between the annotated TIS and the first in-frame upstream stop codon, and a putative tryptic peptide was added to the database for each one after replacing the first amino acid (for non-canonical start codons) with methionine. A similar scan was performed for alternative downstream TIS, limited to a maximum of four.

These three additional databases were combined with databases of the annotated RB-1B proteins, the annotated host proteins from chicken genome assembly bGalGal1.mat.broiler. GRCg7b, and the cRAP database of common contaminant proteins (https://www.thegpm.org/crap/), along with reversed decoy sequences of each entry. Raw spectra were searched against this database using Comet v. 2019.01 rev. 5 [90], MS-GF+ v. 2022.01.07 [91], and Byonic as described above. Search parameters included a precursor mass tolerance of 7 ppm; high-resolution MS2 mass tolerance (MS-GF+ InstrumentID = 1, Comet fragment_bin_tol = 0.02 + fragment_bin_offset = 0.0); fully tryptic termini; maximum two missed cleavages; fixed Cys carbamidomethylation; variable S/T/Y phosphorylation, Met oxidation, N/Q deamidation, N-terminal protein acetylation, and N-terminal methionine excision. Raw spectral hits were post-processed using Percolator v. 3.05 [92] to assign q-values at the spectrum, peptide, and protein levels for use in false discovery rate (FDR) filtering. Comet and Percolator were run within the Crux toolkit v. 4.1 [93]. Visualization of identified peptides was performed in IGV [94]. All search databases and Crux and MS-GF+ configuration files are available upon request.

Peptide intensity calculation was performed using FlashLFQ v. 1.2.4 [95] with match-between-run (MBR) enabled, inter-sample normalization, and requiring MS2 ID in condition for MBR. Peptide intensities were used to calculate protein iBAQ values by dividing summed peptide intensities for each protein by the number of theoretical fully tryptic peptides length 6–40 in the protein. Intensities for peptides shared between proteins were divided evenly between proteins. Relative iBAQ (riBAQ) was calculated within each replicate as the protein iBAQ divided by the sum of iBAQ values for the replicate, considering only viral proteins.

### Data visualization and statistical analysis

Genome tracks and additional data tracks generated from this study were visualized using desktop IGV [94]. Online visualization of the data tracks was built using igv.js [96]. Sequence conservation logos were generated with WebLogo v3.7.12. [97] Static genome maps (Figs 2–6) were built using the R package Gviz v1.38.0 [98]. Most figures were generated using either ggplot2 or BioRender.com. Statistical analysis of the RNA-Seq and proteomics data was performed using the R software package v. 4.1.3 [99]. Analysis of proteomics data within R was partially performed using the MSnbase package v. 2.20.4 [100]. Visualizations of annotated MS2 spectra and combined extracted ion chromatograms (XICs) were created using ms-perl (https://metacpan.org/pod/MS) and R. For the purpose of six-replicate XICs, the raw data were aligned across retention times using the MapAlignerPoseClustering tool from OpenMS [101], with superimposer:mz_pair_max_distance = 0.05 and pairfinder:distance_MZ: max_difference = 7 ppm.

### Supporting information

**S1 Fig. Unique peptide counts and breadth of coverage for detected viral proteins in infected epithelial skin cells.** (A) Frequency of proteins by the number of unique peptides identified from them. (B) Frequency of proteins by their amino acid coverage, defined as the

proportion of amino acids in the sequence likely to be detected by MS/MS (tryptic peptides $\geq$ 6 aa) that are covered by detected peptides.
(TIF)

**S2 Fig. Evidence of expression of MDV023 (pUL11) in infected cells based on a single peptide.** (A) Protein sequence of MDV023 (pUL11) comparing the reference sequence (G9CUB8) and the RB-1B strain used in this study, plus the predicted tryptic cleavage sites. (B) Representative MS2 spectra with annotated b/y ion series, and XIC elution profiles of the peptide mass in replicates of mock- and infected samples.
(TIF)

**S3 Fig. Evidence for expression of MDV064 (pUL49.5-gN) in infected cells based on a single peptide.** (A) Protein sequence of MDV064 (pUL49.5-gN) comparing the reference sequence (QM77MR4) and the RB-1B strain used in this study, plus the predicted tryptic cleavage sites, predicted signal peptide and transmembrane regions. (B) Representative MS2 spectra with annotated b/y ion series, and XIC elution profiles of the peptide mass in replicates of mock- and infected samples.
(TIF)

**S4 Fig. Annotated MS2 spectra and XIC profiles of single peptides for MDV096 (Meq), MDV094 (SORF4), and MDV091.5.** Representative MS2 spectra with annotated b/y ion series, and XIC elution profiles of the peptide mass in replicates of mock- and infected samples for MDV096 (A), MDV094 (B), and MDV0915 (C), indicating poor support for their correct identifications.
(TIF)

**S5 Fig. Genome tracks and short-read RNA-Seq coverage plots for representative regions of the chicken genome (A, B) and viral genome (C, D).**
(TIF)

**S6 Fig. Baseline read depth distribution calculated for intergenic/antisense regions.** Shown are kernel density estimates for read coverage on forward (green) and reverse (red) strands calculated for each genomic position in assumed intergenic and antisense regions. A Gaussian distribution was fit to the data with mean = 50 and SD = 33 (blue). These values were used as estimates of background read coverage to calculate the threshold for true expression (mean + 2 SD).
(TIF)

**S7 Fig. Correlation between viral RNA and protein levels epithelial skin cells.** Shown is a scatterplot of log2-transformed median short-read RNA-Seq read depth and log2 relative protein abundance (from riBAQ calculation) for each protein detected by both methods. Outliers are labeled with gene identifiers.
(TIF)

**S8 Fig. IsoSeq saturation curve.** Shown is the saturation curve from SQANTI3 rarefaction analysis, with isoform count as a function of read depth. The vertical black line shows the actual sequencing depth of the experiment, and the points to the right of this line are extrapolated. FSM = full splice match; NIC = novel in catalog; NNC = novel not in catalog.
(TIF)

**S9 Fig. Distribution of IsoSeq transcript assigned read counts.** Counts represent the sum of FLNC reads assigned to each transcript by the *isoseq3* software.
(TIF)

**S10 Fig. Distribution of 132bp tandem direct repeat copy numbers in IsoSeq FLNC reads.** Long reads overlapping the 132 bp repeat region were extracted, and the repeat copy number for each was tabulated. Copy numbers in the plot are expressed relative to the three copies present in the reference genome (e.g., +1 represents four copies, -1 represents two copies).
(TIF)

**S11 Fig. Example TSS peak visualization.** Shown is a representative region of the MDV genome in the IGV browser, with TSS peak tracks (+/- strands) as well as the annotated coding sequence and predicted transcript models from *isoseq3*, demonstrating the relative inaccuracy of the isoseq3 transcript 5' ends under the settings used as compared with actual read start coverage (TSS peaks) at each position.
(TIF)

**S12 Fig. Distribution of 5' UTR lengths of MDV transcripts in epithelial skin cells.** 5'UTRs were determined for all genes assigned primary TSS, and the length calculation accounted for introns within the 5' UTRs of MDV008, MDV018, MDV037.
(TIF)

**S13 Fig. Sequence conservation in the viral TSS genomic context.** Shown are the sequence logos for the genomic sequence ± 5 bp of the (A) major TSS (as determined from IsoSeq read alignments), (B) secondary TSS, and (C) major + secondary TSS which do not contain a pyrimidine-purine dinucleotide at the TSS.
(TIF)

**S14 Fig. Viral polyadenylation signals.** (A) Distance from the site of cleavage/polyadenylation (as determined by IsoSeq) to the upstream polyadenylation signal hexamer. (B) Sequence conservation logo for the surrounding genomic context centered on the cleavage/polyadenylation site. The plot is numbered from the cleavage location as determined by FLNC read termination, but the observed site is likely to be 1 bp upstream of the actual site as a result of the terminal adenine being removed by the IsoSeq processing software during poly-A tail removal. The upstream region of the conserved hexamer motif and downstream U(T)-rich region are clearly visible.
(TIF)

**S15 Fig. Rich b/y ion series and infection-specific elution profiles of exon junction-spanning peptides for (A) MDV078/vCXCL13, (B) MDV057.1/gC104 (B), and (C) MDV075/14 kDa A (C).** Shown for each peptide are MS2 spectra for the top PSMs annotated with b/y ion series, as well as aligned XIC elution profiles for the peptide mass for each of the six replicates. Each peptide is found only in infected replicates, as indicated by the XIC plots.
(TIF)

**S16 Fig. Poor b/y ion series and/or non-specific elution profiles of peptides.** Shown for each peptide are MS2 spectra for the top PSMs annotated with b/y ion series, as well as aligned XIC elution profiles for the peptide mass for each of the six replicates. Each peptide is found only in infected replicates, as indicated by the XIC plots. These peptide identifications are poorly supported due to a combination of weak ion series and/or XICs showing the precursor mass eluting in both infected and mock replicates.
(TIF)

**S17 Fig. Alternative TIS for MDV055 (pUL42).** (A) 5' end of MDV055 showing N-terminal peptides for both the annotated and alternative TIS identified by peptides. (B & C) For each

peptide, annotated MS2 spectra for the top PSMs and six-replicate aligned XICs are shown.
(TIF)

**S18 Fig. Alternative TIS for MDV070 (pUL55).** (A) 5' end of MDV070 showing N-terminal peptides for both the annotated and alternative TIS identified by peptides. (B & C) For each peptide, annotated MS2 spectra for the top PSMs and six-replicate aligned XICs are shown. (TIF)

**S19 Fig. Alternative splicing of MDV008/MDV073.** (A) MDV008 and MDV073 overlap the junction between the UL and RL regions, creating alternative proteins including previously identified pp38 and pp24, and pp38A and pp38B created through alternative splicing. A novel splice variant termed Novel pp38C is expressed in epithelial skin cells. Donor ("D") and acceptor ("A") locations are shown. (B) MUSCLE alignment of pp38A, pp38B, and Novel pp38C with trypsin cleavage sites. (C) MUSCLE alignment of pp38, pp24, pp38A, pp38B, and Novel pp38C and peptides detected in epithelial skin cells. Some peptides are unique to specific proteins.
(TIF)

**S20 Fig. Annotated MS2 spectra and elution profiles for peptides matching MDV075.2.**
(TIF)

**S21 Fig. Annotated MS2 spectra and elution profiles for the exon-spanning peptide of the novel 14 kDa D isoform.**
(TIF)

**S22 Fig. Tryptic map and elution profiles for SORF6.** (A) Exon I and II of SORF6, location of tryptic cleavage sites, and peptide identified in infected samples. (B &C) MS2 spectra and six-replicate XIC profiles for the putative SORF6 peptide (the two spectra are assigned to an identical peptide with a phosphorylation localized to different serine residues).
(TIF)

**S23 Fig. Novel microORF within MDV060.** (A) 5' region of MDV060 with the coding sequence, TIS for pUL47, and identified peptides mapping to annotated pUL47 in green. A novel peptide (blue) was detected using 6-frame translation search mapping to a novel out-of-frame 5' microORF. Two potential TIS for the novel microORF are shown. (B) For the novel peptide, annotated MS2 scans from the top PSMs showing rich y-ion series and six-replicate XIC showing precursor specificity for infected replicates.
(TIF)

**S24 Fig. Novel microORF within MDV084 (ICP4).** (A) 5' region of MDV084 with the coding sequence, TIS for ICP4, and identified peptides mapping to annotated ICP4 in green. A novel peptide (blue) was detected using 6-frame translation search mapping to a novel out-of-frame 5' microORF (orange). B) For the novel peptide, annotated MS2 scans from the top PSMs showing decent y-ion series with strong proline-effect peaks and six-replicate XIC showing precursor specificity for infected replicates.
(TIF)

**S25 Fig. Tryptic map for MDV056 (pUL43).** Protein sequence of MDV056 (pUL43) comparing the reference sequence (Q9E6M9) and the RB-1B strain used in this study, plus the predicted tryptic cleavage sites. Transmembrane regions and the unique peptide identified in Liu et al. [23] are shown.
(TIF)

**S26 Fig. Tryptic map for MDV032 (pUL20).** Protein sequence of MDV032 (pUL20) comparing the reference sequence (Q77MS4) and the RB-1B strain used in this study. The predicted tryptic cleavage sites and transmembrane regions are shown.
(TIF)

**S27 Fig. Tryptic map for MDV035 (pUL24).** Protein sequence of MDV035 (pUL24) comparing the reference sequence (Q9E6P4) and the RB-1B strain used in this study. The predicted tryptic cleavage sites and unique peptides identified in Liu et al. [23] are shown.
(TIF)

**S28 Fig. Predicted protein lengths for annotated MDV genes.** Stacked histogram shows distribution of protein lengths grouped by short-read RNA-Seq expression status (see text for criteria used).
(TIF)

**S1 Table. MS/MS summary statistics for unenriched and phospho-enriched protein extracts of three mock- and three MDV-infected samples.**
(XLSX)

**S2 Table. MDV peptides identified in epithelial skin cells at 1% FDR.**
(XLSX)

**S3 Table. Summary of MDV RNA and proteins in epithelial skin cells.** Worksheets show viral genes detected at both RNA and protein level, RNA only, and not detected in either RNA-Seq or LC-MS/MS.
(XLSX)

**S4 Table. Summary of MDV short-read RNA sequencing data in epithelial skin cells.**
(XLSX)

**S5 Table. Summary of MDV long-read IsoSeq RNA sequencing data in epithelial skin cells.**
(XLSX)

**S6 Table. Transcript start sites (TSS) detected by long-read IsoSeq sequencing.**
(XLSX)

**S7 Table. Summary of MDV protein N-termini and associated PTMs detected by MS/MS in epithelial skin cells.**
(XLSX)

**S8 Table. Label-free quantification of MDV peptides detected in epithelial skin cells.**
(XLSX)

**S9 Table. Novel viral peptides detected using six-frame proteogenomic searching.**
(XLSX)

**S10 Table. Comparison of the current and former reports for RNA sequencing and proteomics during MDV infection in different cell systems.**
(DOCX)

**S1 File. Data tracks from study used for visualization and the web-based genome browser.** A table of file descriptions is included in the archive.
(ZIP)

## Acknowledgments

We thank the Georgia Genomics and Bioinformatics Core, which provided the Illumina: Ribo-depleted RNA library preparation and NextSeq500 2×75bp sequencing service and the University of Illinois at Urbana-Champaign Proteomics Core for their guidance in LC-MS/MS services.

## Author Contributions

**Conceptualization:** Stephen J. Spatz, Keith W. Jarosinski.

**Data curation:** Jeremy D. Volkening, Nagendraprabhu Ponnuraj, Haji Akbar, Justine V. Arrington.

**Formal analysis:** Jeremy D. Volkening, Stephen J. Spatz, Justine V. Arrington, Keith W. Jarosinski.

**Funding acquisition:** Stephen J. Spatz, Keith W. Jarosinski.

**Investigation:** Jeremy D. Volkening, Nagendraprabhu Ponnuraj, Haji Akbar, Justine V. Arrington, Widaliz Vega-Rodriguez, Keith W. Jarosinski.

**Methodology:** Stephen J. Spatz, Nagendraprabhu Ponnuraj, Haji Akbar, Justine V. Arrington, Widaliz Vega-Rodriguez.

**Project administration:** Keith W. Jarosinski.

**Resources:** Stephen J. Spatz, Justine V. Arrington.

**Software:** Jeremy D. Volkening, Justine V. Arrington.

**Supervision:** Keith W. Jarosinski.

**Validation:** Justine V. Arrington.

**Visualization:** Keith W. Jarosinski.

**Writing – original draft:** Jeremy D. Volkening, Keith W. Jarosinski.

**Writing – review & editing:** Jeremy D. Volkening, Stephen J. Spatz, Nagendraprabhu Ponnuraj, Haji Akbar, Justine V. Arrington, Keith W. Jarosinski.

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
