## [Decision Letter · Decision Letter 0]

13 Mar 2023

Dear Professor Jarosinski,

Thank you very much for submitting your manuscript "Viral proteogenomic and expression profiling during fully productive replication of a skin-tropic herpesvirus in the natural host" for consideration at PLOS Pathogens. As with all papers reviewed by the journal, your manuscript was reviewed by members of the editorial board and by several independent reviewers. While each of the reviewers were impressed with the scope of work done, they also pointed out several weaknesses in the current version.  This includes the perception of a lack of novelty in the approach and the descriptive nature of the studies presented.  Reviewers suggested that some of these weaknesses could be mitigated with a more extensive and in depth re-write.  Reviewers also noted that two additional experiments would further strengthen the manuscript experimentally and should be included.  Please be sure to address all of the reviewers comments in the revised version of the manuscript.

We cannot make any decision about publication until we have seen the revised manuscript and your response to the reviewers' comments. Your revised manuscript is also likely to be sent to reviewers for further evaluation.

Sincerely,

Donna M Neumann

Academic Editor

PLOS Pathogens

Alison McBride

Section Editor

PLOS Pathogens

Kasturi Haldar

Editor-in-Chief

PLOS Pathogens

orcid.org/0000-0001-5065-158X

Michael Malim

Editor-in-Chief

PLOS Pathogens

orcid.org/0000-0002-7699-2064

Reviewer's Responses to Questions

**Part I - Summary**

Reviewer #1: This paper by Volkening et al. describes the proteome and stable transcriptome of Marek’s disease virus (MDV) in the skin of chickens. The skin is of interest here because it is the only place in the animal that allows production of infectious virus despite systemic infection. In this way it is reminiscent of Varicella Zoster Virus, another alpha herpesvirus, that replicates in skin cells but is otherwise restricted.

Regarding the significance of the study, there have been several -omic type descriptive papers on MDV recently focusing on either RNA or protein in cultured cells or B cells, but this is the first to combine these in a single set of analyses in a cell type fully permissive to infectious virus production in vivo. The data is extensive, well presented, and will likely represent the go-to data set for MDV gene and protein expression. Although there were relatively few surprises beyond what is expected from annotated genes some areas of interest that have been confusing from the annotation are clarified, including alternative splicing in the abundantly expressed gC and 14Kda loci.

Given the lack of surprises and what appear to be largely similar results from the previous studies, the novelty of the data is not particularly high, although the studies are performed thoroughly and data curated and presented well. A comparative table might be useful to highlight results that are different between the current and previous work.

Reviewer #2: In this manuscript, the authors assessed the transcriptome and proteome of the highly oncogenic Marek’s disease virus (MDV) in feather follicle epithelia cells (FFE). The FFE is particular important as this is the only site in animals, where infectious cell free MDV is produced and shed into the environment, which ensure the spread of the virus. Until now, both the transcriptome and proteome in this crucial site remained elusive. This study provides a comprehensive overview and includes a plethora of information that are exciting to the field (as well as other alphaherpesviruses such as VZV) and will provide the basis for many future studies. Overall, the manuscript is well written but is also very technical due to the nature of transcriptome and proteome analyses. Only the following points should be addressed prior to publication.

Reviewer #3: Volkening et al have generated detailed maps of the transcriptome and proteome of Marek’s disease virus (MDV) in feather follicle epithelial skin cells of live chickens. Their powerful approach provides the ability to link viral transcription and protein production in the most relevant system available (live chickens). The analysis presented is thorough and without doubt this work further enhances our knowledge of MDV biology. However, while the level of detail is high, the overarching purpose and results of this study are somewhat lost in the writing (i.e. it is hard to see the wood for the trees). It is thus a very descriptive work which lacks wider context. There also appears to be some missed opportunities to obtain data that would really push this work to the level it deserves. With that in mind, I offer the following critiques and recommended revisions which I consider essential for this work to be accepted.

**Part II – Major Issues: Key Experiments Required for Acceptance**

Reviewer #1: Data is thorough.

Reviewer #2: (No Response)

Reviewer #3: 1. On reading the abstract and introduction, I remain uncertain as to the purpose of the study. While there is obvious value in looking at the viral transcriptome and proteome in this system, the authors do not convey this well, nor do they provide any real comparative analysis against cell culture models to show why their system is so valuable.

2. Numerous recent studies have demonstrated the major advantages of long-read sequencing approaches in generate ultra-high resolution maps of viral transcriptomes. While input requirements for some of these (e.g. PacBio, Nanopore Direct RNA-Sequencing, Direct cDNA-Sequencing) are problematic for non-cell culture based systems, approaches such as Nanopore PCR-cDNA sequencing should be compatible. Alternatively, the authors could apply a long-read approach to a cell culture based system to at least characterize the full diversity of the MDV transcriptome and then integrate the RNA-Seq and proteome data from the feather follicle system to look at differences in expression level. While the optimal approach would be to include long-read sequencing datasets for both the feather follicle and a cell culture based system, even a single dataset from the latter could be sufficient to provide a much more detailed analysis including the identification of the boundaries of all encoded mRNAs. This would also significantly enhance the global analysis of splicing patterns across the MDV transcriptome.

3. From a transcription perspective, figures such as S1-S4 would be valuable to include within the main text itself. It is also confusing to see tracks named as ‘genes’ and ‘introns’ rather than having transcript structures (e.g. mRNAs) represented. Similarly, the use of a log2 scale on the coverage plots actually makes it harder to discern regions with abundant transcription from regions with low-level transcription.

4. The observation regarding lower strand-specificity of the reads aligning to the viral genome when compared to the host bears further investigation. It is not clear from the results/methods how this value was calculated but it seems likely to be an artefact of the analysis rather than a real biological phenomenon. Moreover, the calculation might be biased by the decision to remove terminal repeats from the genome (as specified in the methods). The authors should consider determining the strand-specificity values for the UL and US regions alone to determine whether the observed 66-75% is consistent across the genome. Finally, the analysis of the high strand-specificity observed on the host genome should be shown (rather than just reported as unpublished observation).

5. The observation that the most abundant transcript (B68) is not obviously translated is very interesting. There are various softwares available that can predict whether an RNA is likely to be coding or non-coding. Further analysis, potentially including some in situ hybridization to see where the RNA localizes would seem warranted here.

6. The authors might discuss other approaches that could be used for orthogonal validation (e.g. ribosome profiling) of their results and why these have not been used. They might also point out that their approach is generally rather conservative. This is not a criticism per se but it does warrant some mention as there are sensitivity/specificity issues with both the RNA-Seq and MS approaches used.

7. There remains an overall lack of context in this study. The authors mention at least three other recent efforts to profile/annotate the MDV transcriptome but does not discuss the data presented here relative to those. While a comprehensive comparison is probably beyond the scope of this study, it would still be useful to have some discussion of similarities and differences. In studies of other alphaherpesviruses, it has generally been seen that while expression levels of viral transcripts and proteins may differ between cell types, viral strains, and experimental models, the breadth of the transcriptome (and thus the inferred proteome) remains consistent. If that is not the case for MDV then that would constitute a significant and interesting finding, particularly if it can be demonstrated that novel transcripts/proteins are expressed in live infected chickens that have never been seen in cell culture models.

**Part III – Minor Issues: Editorial and Data Presentation Modifications**

Reviewer #1: A comparative table might be useful to highlight results that are different between the current and previous work.

Reviewer #2: 1) The supplementary figures contain a lot of valuable data and provide a good overview of the transcriptome, peptide, splice variants aso. Unfortunately, the resolution of the supplementary figures S1-4, S6-8 and S11-20 is very low and many detail are poorly visible or blurry. As this information is important, the authors must be improved the resolution of these figures.

2) When the authors describe the recombinant virus in the Material and Methods, they must include the information on the strain (RB-1B). It is nice to know the modifications made in the virus genome, but the strain is even more important.

3) The authors state that they infected 12 chickens and kept 14 uninfected animals. Were samples pooled from several chickens to obtain the 3 and 6 replicates for MS and RNA-Seq respectively? Or were not all of these chickens used in the study? Please provide more info on this aspect.

4) Line 20: “little is known about the viral genes that mediate transmission, mostly due to their close relationship to their natural host.” It is not clear what the author means with close relationship. Please clearly phrase this sentence.

5) Line 56: the statement that “most herpesviruses are primarily cell-associated in cell culture and within the host” is not correct. Most alphaherpesviruses efficiently spread via cell free virions. Please correct…

Reviewer #3: Line 24: The authors should more clearly define ‘fully productive replication’ - I think they mean this results in virus being shed outside of the host but they language is somewhat confusing as the virus can also replicate and spread between other cells (which might be considered by some as fully productive replication).

Line 69: The alphaherpesvirus family is very large with only a few members well studied. The authors should clarify that ‘of the well studied herpesviruses, MDV is the only one which appears strictly cell-associated’

Line 79: Relevant studies should be cited here.

Table S4: The fraction column (I) shows 0 for all datasets when this is obviously not true. The authors should also include the fraction of reads aligning to the host genome and the fraction that aligns to neither host nor virus.

Methods: The minimum quality score applied during TrimGalore trimming of the RNA-Seq data appears very low (8?). This could lead to a number of reads with low quality 3’ ends that fail to align and a loss of data. The authors should consider higher quality values (20 and 30) to see whether this improves the fraction of reads aligning and thus the overall quality and depth of the data.

PLOS authors have the option to publish the peer review history of their article (what does this mean?). If published, this will include your full peer review and any attached files.

Reviewer #1: No

Reviewer #2: No

Reviewer #3: No

Figure Files:

Data Requirements:

Please note that, as a condition of publication, PLOS' data policy requires that you make available all data used to draw the conclusions outlined in your manuscript. Data must be deposited in an appropriate repository, included within the body of the manuscript, or uploaded as supporting information. This includes all numerical values that were used to generate graphs, histograms etc.. For an example see here on PLOS Biology: http://www.plosbiology.org/article/info:doi%2F10.1371%2Fjournal.pbio.1001908#s5.
---

## [Editor Report · Decision Letter 1]

29 May 2023

Dear Dr. Jarosinski

We are pleased to inform you that your manuscript 'Viral proteogenomic and expression profiling during productive replication of a skin-tropic herpesvirus in the natural host' has been provisionally accepted for publication in PLOS Pathogens.

Best regards,

Donna M Neumann

Academic Editor

PLOS Pathogens

Alison McBride

Section Editor

PLOS Pathogens

Kasturi Haldar

Editor-in-Chief

PLOS Pathogens

orcid.org/0000-0001-5065-158X

Michael Malim

Editor-in-Chief

PLOS Pathogens

orcid.org/0000-0002-7699-2064
---

## [Editor Report · Acceptance letter]

7 Jun 2023

Dear Dr. Jarosinski,

We are delighted to inform you that your manuscript, "Viral proteogenomic and expression profiling during productive replication of a skin-tropic herpesvirus in the natural host," has been formally accepted for publication in PLOS Pathogens.

Best regards,

Kasturi Haldar

Editor-in-Chief

PLOS Pathogens

orcid.org/0000-0001-5065-158X

Michael Malim

Editor-in-Chief

PLOS Pathogens

orcid.org/0000-0002-7699-2064